# Learning to Generate Noise for Multi-Attack Robustness

## Abstract

Adversarial learning has emerged as one of the successful techniques to circumvent the susceptibility of existing methods against adversarial perturbations. However, the majority of existing defense methods are tailored to defend against a single category of adversarial perturbation (e.g. $\ell_\infty$-attack). In safety-critical applications, this makes these methods extraneous as the attacker can adopt diverse adversaries to deceive the system. Moreover, training on multiple perturbations simultaneously significantly increases the computational overhead during training. To address these challenges, we propose a novel meta-learning framework that explicitly learns to generate noise to improve the model's robustness against multiple types of attacks. Its key component is *Meta Noise Generator (MNG)* that outputs optimal noise to stochastically perturb a given sample, such that it helps lower the error on diverse adversarial perturbations. By utilizing samples generated by MNG, we train a model by enforcing the label consistency across multiple perturbations. We validate the robustness of models trained by our scheme on various datasets and against a wide variety of perturbations, demonstrating that it significantly outperforms the baselines across multiple perturbations with a marginal computational cost.

## 1 Introduction

Deep neural networks have demonstrated enormous success on multiple benchmark applications (Amodei et al., 2016; Devlin et al., 2018), by achieving super-human performance on certain tasks. However, to deploy them to safety-critical applications (Shen et al., 2017; Chen et al., 2015; Mao et al., 2019), we need to ensure that the model is *robust* as well as *accurate*, since incorrect predictions may lead to severe consequences. Notably, it is well-known that the existing neural networks are highly susceptible to carefully crafted image perturbations which are imperceptible to humans but derail the predictions of these otherwise accurate networks.

The emergence of adversarial examples has received significant attention in the research community, and several defense mechanisms have been proposed (Madry et al., 2017; Zhang et al., 2019; Carmon et al., 2019). However, despite a large literature to improve upon the robustness of neural networks, most of the existing defenses leverage the knowledge of the adversaries and are based on the assumption of only a single type of perturbation. Consequently, many of the proposed defenses were circumvented by stronger attacks (Athalye et al., 2018; Uesato et al., 2018; Tramer et al., 2020).

Meanwhile, several recent works have (Schott et al., 2018; Tramèr & Boneh, 2019) demonstrated the vulnerability of existing defense methods against multiple perturbations. For the desired multi-attack robustness, Tramèr & Boneh (2019); Maini et al. (2020) proposed various strategies to aggregate multiple perturbations during training. However, training with multiple perturbations comes at an additional cost; it increases the training cost by a factor of four over adversarial training, which is already an order of magnitude more costly than standard training. This slowdown factor hinders the research progress of robustness against multiple perturbations due to the large computation overhead incurred during training. Some recent works reduce this cost by reducing the complexity of generating adversarial examples (Shafahi et al., 2019; Wong et al., 2020), however, they are limited to $\ell_\infty$ adversarial training.

To address the drawbacks of existing methods, we propose a novel training scheme, *Meta Noise Generator with Adversarial Consistency (MNG-AC)*, which learns instance-dependent noise to

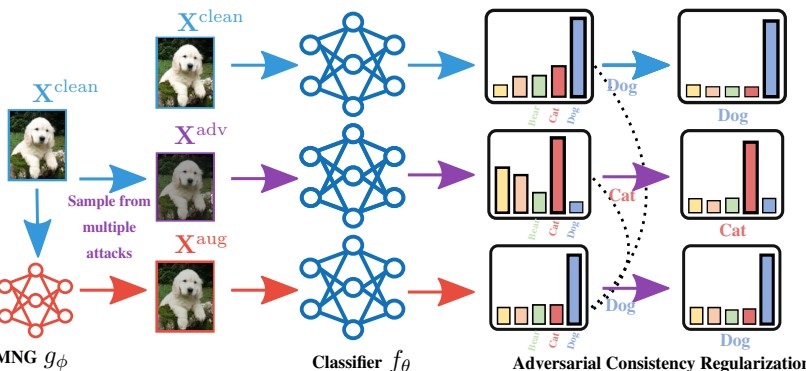

Figure 1: **Overview of Meta-Noise Generator with Adversarial Consistency (MNG-AC).** First, we stochastically sample a perturbation to generate the adversarial examples $X^{\mathrm{adv}}$. The generator $g_\phi$ takes stochastic noise and input $X^{\mathrm{clean}}$ to generate the noise-augmented sample $X^{\mathrm{aug}}$. The classifier $f_\theta$ then minimizes the stochastic adversarial classification loss and the adversarial consistency loss. MNG is learned via meta-learning to explicitly minimize the adversarial classification loss.

minimize the adversarial loss across multiple perturbations while enforcing label consistency between them, as illustrated in Figure 1 and explained in details below.

First, we tackle the heavy computational overhead incurred by multi-perturbation training by proposing *Stochastic Adversarial Training (SAT)*, that samples from a distribution of perturbations during training, which significantly accelerates training for multiple perturbations[1]. Then, based on the assumption that the model should output the same predictions for different perturbations of the same image, we introduce *Adversarial Consistency (AC)* loss that enforces label consistency across multiple perturbations. Finally, motivated by the noise regularization techniques (Huang et al., 2016; Srivastava et al., 2014; Noh et al., 2017; Lee et al., 2020) which target generalization, we formulate a *Meta Noise Generator (MNG)* that learns to stochastically perturb a given sample in a meta-learning framework to explicitly improve the generalization and label consistency across multiple attacks. In particular, MNG-AC utilizes our generated samples to enforce label consistency across the generated samples from our model, adversarial samples, and clean samples. Consequently, it pushes the decision boundary (see Figure 4) and enforces a smooth and robust network across multiple perturbations.

We validate the efficacy and efficiency of our proposed method by comparing it against existing, state-of-the-art methods on CIFAR-10, SVHN, and Tiny-ImageNet dataset. The experimental results show that our method obtains significantly superior performance over all the baseline methods trained with multiple perturbations, generalizes to diverse perturbations, and substantially reduces the computational cost incurred by training with multiple perturbations. In summary, the major contributions of this paper are as follows:

- We introduce *Adversarial Consistency (AC)* loss that enforces label consistency across multiple perturbations to enforce smooth and robust networks.

- We formulate *Meta-Noise Generator* that explicitly meta-learns an input-dependent noise generator, such that it outputs stochastic noise distribution to improve the model's robustness and adversarial consistency across multiple types of adversarial perturbations.

- We validate our proposed method on various datasets against diverse benchmark adversarial attacks, on which it achieves state-of-the-art performance, highlighting its practical impact.

## 2 RELATED WORK

**Robustness against single adversarial perturbation.** In the past few years, multiple defenses have been proposed to defend against a single type of attack (Madry et al., 2017; Xiao et al., 2020; Zhang et al., 2019; Carmon et al., 2019) and have been consequently circumvented by subsequent attacks (Athalye et al., 2018; Brendel et al., 2018; Tramer et al., 2020). Adversarial-training based

---

[1]By a factor of four on a single machine with four GeForce RTX 2080Ti on CIFAR-10 and SVHN dataset using Wide ResNet 28-10 (Zagoruyko & Komodakis, 2016) architecture.

defenses (Madry et al., 2017; Zhang et al., 2019; Carmon et al., 2019) have been the only exceptions that have withstood the intense scrutiny and have provided empirical gains in adversarial robustness.

**Generative models for adversarial robustness.** There have been various attempts that leverage the representative power of generative models to improve model robustness. Samangouei et al. (2018); Jalal et al. (2017) project an image onto the generator manifold, which is then classified by the discriminator. Song et al. (2018) uses the sensitivity of generative models to defend against a single perturbation. Yin et al. (2020) proposed a detection method based on input space partitioning. However, Samangouei et al. (2018); Jalal et al. (2017); Song et al. (2018) were shown to be ineffective by stronger attacks (Carlini & Wagner, 2017; Athalye et al., 2018). In contrast to learning the generative model to model the adversarial examples, we meta-learn the generator to explicitly learn an input-dependent optimal noise distribution to lower adversarial error across multiple perturbations, that does not necessarily correspond to any of the attack perturbations.

**Robustness against multiple adversarial perturbations.** Schott et al. (2018) demonstrated that $\ell_\infty$ adversarial training is highly susceptible to $\ell_0/\ell_2$-norm adversarial perturbations and used multiple VAEs to defend against multiple perturbations on the MNIST dataset. However, it was not scalable and limited to the MNIST dataset. Tramèr & Boneh (2019) investigated the theoretical/empirical trade-offs between multiple perturbations and introduced adversarial training with worst/average perturbations to defend against multiple perturbations. Maini et al. (2020) incorporated multiple perturbations into a single adversary to maximize the adversarial loss. However, computing all the perturbations is impractical for multiple perturbations and large scale datasets. On the other hand, our proposed framework overcomes this limitation, with improved performance over these methods and has a negligible increase in training cost over multi-perturbation adversarial training.

## 3 ROBUSTNESS AGAINST MULTIPLE PERTURBATIONS

We first briefly review single/multi-perturbation adversarial training and introduce *Stochastic Adversarial Training (SAT)* to reduce the computational cost incurred by training with multiple perturbations. We consider a dataset $\mathcal{D}$ over observations $x \in \mathbb{R}^d$ and labels $y \in \mathbb{R}^C$ with $C$ classes. Let $f_\theta : \mathbb{R}^d \to \mathbb{R}^C$ be a $L$-layered classifier with parameters $\theta$ and classification loss $\mathcal{L}_{\text{cls}}$. Given an attack procedure $\mathcal{A}(x)$ with norm-ball $\mathcal{B}_\mathcal{A}(x, \varepsilon)$ around $x$ with radius $\varepsilon$ for each example, which introduces a perturbation $\delta$, we let $x^{\text{adv}} = x + \delta$ denote the corresponding adversarial examples. We consider the $\ell_p$ norm distance under the additive threat model (Laidlaw & Feizi, 2019) and adopt the projected-gradient descent (PGD) (Madry et al., 2017) for crafting the $\ell_p$ perturbations:

$$x^{\text{adv}}_{(t+1)} = \underset{\mathcal{B}_\mathcal{A}(x, \varepsilon)}{\text{proj}} \left( x^{\text{adv}}_{(t)} + \underset{||v||_\mathcal{A} \leq \alpha_\mathcal{A}}{\arg\max} \, v_\mathcal{A}^T \nabla_{x^{\text{adv}}_{(t)}} \mathcal{L}_{\text{cls}} \left( f_\theta \left( x^{\text{adv}}_{(t)} \right), y \right) \right), \tag{1}$$

where $x^{\text{adv}}_0$ is chosen at random within $\mathcal{B}_\mathcal{A}(x, \varepsilon)$, $\alpha_\mathcal{A}$ is the step size, $\text{proj}$ is the projection operator projecting the input onto the norm ball $\mathcal{B}_\mathcal{A}(x, \varepsilon)$, and $x^{\text{adv}}_{(t+1)}$ denotes the adversarial example at the $t$-th PGD step. We will refer the approximation of the maximum loss by an attack procedure $\mathcal{A}(x)$, such that $\max_{\delta \in \mathcal{B}_\mathcal{A}(x, \varepsilon)} \mathcal{L}_{\text{cls}} \left( f_\theta \left( x + \delta \right), y \right) \approx \mathcal{L}_{\text{cls}} \left( f_\theta \left( \mathcal{A}(x) \right), y \right)$ for the rest of our paper.

**Single-perturbation adversarial training.** In the standard single-perturbation adversarial training (Kurakin et al., 2016; Madry et al., 2017), the model optimizes the network using a min-max formulation. More formally, the inner maximization generates the adversarial perturbation by maximizing the loss, while the outer minimization minimizes the loss on the generated examples.

$$\min_\theta \, \mathbb{E}_{(x,y) \sim \mathcal{D}} \, \mathcal{L}_{\text{cls}} \left( f_\theta \left( \mathcal{A}(x) \right), y \right). \tag{2}$$

The majority of existing single-perturbation defenses are primarily able to defend against a single category of adversarial perturbation. However, this limits the generalization of these methods to perturbations that are unseen during training (Schott et al., 2018; Tramèr & Boneh, 2019), which has been referred to as *overfitting* on the particular type of training perturbation.

**Multi-perturbation adversarial training.** Tramèr & Boneh (2019) extended the adversarial training to multiple perturbations by optimizing the outer objective in Eq. (2) on the strongest/union of adversarial perturbations for each input example. Their proposed strategies can more formally be defined as follows:

1. **The maximum over all perturbations**: It optimizes the outer objective in Eq. (2) on the strongest adversarial perturbation from the whole set of additive adversarial perturbations

$$\min_\theta \ \mathbb{E}_{(x,y)\sim\mathcal{D}}\left[\arg\max_k \mathcal{L}_{\mathrm{cls}}\left(f_\theta\left(\mathcal{A}_k\left(x\right)\right),y\right)\right]. \tag{3}$$

2. **The average over all perturbations:** It optimizes the outer objective in Eq. (2) on the whole set of $n$ additive perturbations.

$$\min_\theta \ \mathbb{E}_{(x,y)\sim\mathcal{D}}\frac{1}{n}\sum_{k=1}^{k=n}\mathcal{L}_{\mathrm{cls}}\left(f_\theta\left(\mathcal{A}_k\left(x\right),y\right)\right). \tag{4}$$

Recently, Maini et al. (2020) proposed "Multi Steepest Descent" (MSD) by incorporating the different perturbations into the direction of steepest descent. However, the practicality of all these methods is limited due to an increased computational overhead for training.

**Stochastic Adversarial Training (SAT).** To overcome this limitation, we propose Stochastic Adversarial Training to defend against multiple adversarial perturbations. Specifically, we conjecture that it is essential to cover the threat model during training, not utilizing all the perturbations simultaneously. We formulate the threat model as a random attack $\mathcal{A}(x)$ sampled uniformly from a perturbation set $S$ during each episode (or batch) of training which prevents overfitting on a particular adversarial perturbation. In this work, we consider the $\ell_p$-bounded perturbation set, and we sample the attack procedure $\mathcal{A}(x)$ with its corresponding norm-ball $\mathcal{B}_\mathcal{A}(x,\varepsilon)$ from the perturbation set $S$ as follows:

$$S = \{\mathcal{A}_1(x),\ldots,\mathcal{A}_n(x)\},$$
$$k \sim \mathrm{Cat}\left((1/n,\ldots,1/n)\right),$$
$$\mathcal{A}(x) = S_k(x), \tag{5}$$

where $\mathrm{Cat}$ is the categorical distribution and $n$ is the number of attacks in the perturbation set $S$. Our proposed SAT optimizes the outer objective in Eq. (2) using the sampled attack procedure $\mathcal{A}(x)$ and is a drastic simplification of the average one in Eq. (4), which makes it highly efficient for multiple perturbations. It is important to note that unlike the average and max strategy SAT can be applied to any perturbation set with a constant cost and it promotes generalization and convergence (due to its stochasticity) by preventing over-fitting on a single type of perturbation.

## 4    LEARNING TO GENERATE NOISE FOR MULTI-ATTACK ROBUSTNESS

In this section, we introduce our framework MNG-AC, which leverages an *adversarial consistency loss (AC)* and a *meta-noise generator (MNG)* to help the model generalize to multiple perturbations. Let $g_\phi : \mathbb{R}^d \to \mathbb{R}^d$ denote the generator with parameters $\phi$ and $x_\theta^{\mathrm{adv}}$ be the adversarial examples generated by SAT for a uniformly sampled attack $\mathcal{A}(x)$ from a perturbation set $S$ with norm-ball $\mathcal{B}_\mathcal{A}(x,\varepsilon)$. We sample $z \sim \mathcal{N}(0,\mathbf{I})$ for input to our generator jointly with the clean examples $x$ to generate the noise-augmented samples $x_\phi^{\mathrm{aug}}$ projected on the norm-ball $\mathcal{B}_\mathcal{A}(x,\varepsilon)$. Note that, as MNG learns the noise to minimize the adversarial loss, it is essential to project the generated noise on the norm-ball $\mathcal{B}_\mathcal{A}(x,\varepsilon)$, which is the corresponding norm-ball of the sampled attack procedure $\mathcal{A}(x)$. The total loss function $\mathcal{L}_{\mathrm{total}}$ for the classifier consists exclusively of two terms: SAT classification loss and an adversarial consistency loss:

$$\mathcal{L}_{\mathrm{total}} = \frac{1}{B}\sum_{i=1}^{B}\underbrace{\mathcal{L}_{\mathrm{cls}}\left(\theta \mid x_\theta^{\mathrm{adv}}(i),y(i)\right)}_{\text{SAT classification loss}} + \underbrace{\beta\cdot\mathcal{L}_{\mathrm{ac}}\left(p^{\mathrm{clean}}(i);p^{\mathrm{adv}}(i);p^{\mathrm{aug}}(i)\right)}_{\text{adversarial consistency loss}}, \tag{6}$$

where $B$ is the batch-size, $\beta$ is the hyper-parameter determining the strength of the AC loss denoted by $\mathcal{L}_{\mathrm{ac}}$ and $p^{\mathrm{clean}},p^{\mathrm{adv}},p^{\mathrm{aug}}$ represent the posterior distributions $p(y \mid x^{\mathrm{clean}}),p(y \mid x_\theta^{\mathrm{adv}}),p(y \mid x_\phi^{\mathrm{aug}})$ computed using the softmax function on the logits for $x^{\mathrm{clean}}$, $x^{\mathrm{adv}}$, and $x^{\mathrm{aug}}$ respectively. Specifically, $\mathcal{L}_{\mathrm{ac}}$ represents the Jensen-Shannon Divergence (JSD) among the posterior distributions:

$$\mathcal{L}_{\mathrm{ac}} = \frac{1}{3}\left(D_{\mathrm{KL}}(p^{\mathrm{clean}} \parallel M) + D_{\mathrm{KL}}(p^{\mathrm{adv}} \parallel M) + D_{\mathrm{KL}}(p^{\mathrm{aug}} \parallel M)\right), \tag{7}$$

where $M = \left(p^{\mathrm{clean}} + p^{\mathrm{adv}} + p^{\mathrm{aug}}\right)/3$. Consequently, $\mathcal{L}_{\mathrm{ac}}$ enforces stability and insensitivity across a diverse range of inputs based on the assumption that the classifier should output similar predictions when fed perturbed versions of the same image.

---

**Algorithm 1** Learning to generate noise for multi-attack robustness

---

**input** Dataset $\mathcal{D}$, $T$ inner gradient steps, batch size $B$, perturbation set $S$.
**output** Final model paramaters $\theta$
1: **for** $n = \{1, \ldots, N\}$ **do**
2:     Sample mini-batch of size $B$.
3:     Sample an attack procedure $\mathcal{A}(x)$ with its corresponding norm-ball $\mathcal{B}_{\mathcal{A}}(x, \varepsilon)$ using Eq. (5).
4:     Generate the adversarial examples for $\mathcal{A}(x)$ using Eq. (1).
5:     Sample $z \sim \mathcal{N}(0, \mathbf{I})$ and generate $x_{\phi}^{\text{aug}} = \underset{\mathcal{B}_{\mathcal{A}}(x, \varepsilon)}{\text{proj}} (x + g_{\phi}(z, x))$ using MNG, where $\mathcal{B}_{\mathcal{A}}(x, \varepsilon)$
      is the norm-ball corresponding to the attack procedure sampled in Step 3.
6:     Update $\theta$ to minimize Eq. (6).
7:     Initialize $\theta^{(0)} = \theta$
8:     **for** $t = \{1, \ldots, T\}$ **do**
9:         Update $\theta^{(t)}$ using Eq. (8).
10:    **end for**
11:    Descent a single step to update $\theta^{(T)}$ to $\theta^{(T+1)}$ by Eq. (9).
12:    Update the parameters $\phi$ of the generator by Eq. (10).
13: **end for**

---

Recently, Rusak et al. (2020) formulated an adversarial noise generator to learn the adversarial noise to improve the robustness on common corruptions. However, our goal is different; the robustness against multiple adversarial attacks is a much more challenging task than that against common corruptions. To generate the augmented samples for our purpose, we explicitly perturb the input examples for generalization across multiple perturbations. In particular, MNG meta-learns (Thrun & Pratt, 1998; Finn et al., 2017) the parameters $\phi$ of a noise generator $g_{\phi}$ to generate an input-dependent noise distribution to alleviate the issue of generalization across multiple adversaries. The standard approach to train our adversarial classifier jointly with MNG is to use bi-level optimization (Finn et al., 2017). However, bi-level optimization for adversarial training would be computationally expensive.

To tackle this challenge, we adopt an online approximation (Ren et al., 2018; Jang et al., 2019) to update $\theta$ and $\phi$ using a single-optimization loop. We alternatively update the parameters $\theta$ of the classifier with the parameters $\phi$ of MNG. In particular, we first update the parameters $\theta$ using Eq. (6) (step 3 in Algorithm 1). Then, given current parameters $\theta$ of our adversarial classifier, we update MNG parameters $\phi$ using the following training scheme:

1. **Update model parameters for $T$ steps.** First, we update $\theta$ to minimize $\mathcal{L}_{\text{cls}}(\theta \mid x_{\phi}^{\text{aug}}, y)$ for $T$ steps which ensures the learning of the classifier using the knowledge from the generated samples constructed by MNG. It explicitly increases the influence of the noise-augmented samples on the classifier in the inner loop. More specifically, for a learning rate $\alpha$, projection operator $\text{proj}$, $\theta^{(t)}$ moves along the following descent direction on a mini-batch of training data:

$$\theta^{(t+1)} = \theta^{(t)} - \alpha \cdot \frac{1}{B} \sum_{i=1}^{B} \nabla_{\theta} \mathcal{L}_{\text{cls}} \left( \theta^{(t)} \mid x_{\phi}^{\text{aug}}(i), y(i) \right),$$

$$\text{where,} \quad x_{\phi}^{\text{aug}} = \underset{\mathcal{B}(x, \varepsilon)}{\text{proj}} (x + g_{\phi}(z, x)). \tag{8}$$

2. **Adapt model parameters on a single step.** Second, perform one-step update to update $\theta^{(T)}$ to $\theta^{(T+1)}$ to minimize SAT loss from Eq. (6). This step explicitly models the adaptation of adversarial model parameters in the presence of the noise-augmented data using a single step of update:

$$\theta^{(T+1)} = \theta^{(T)} - \alpha \cdot \frac{1}{B} \sum_{i=1}^{B} \nabla_{\theta} \mathcal{L}_{\text{cls}} \left( \theta^{(T)} \mid x_{\theta}^{\text{adv}}(i), y(i) \right). \tag{9}$$

3. **Update generator parameters.** In the last step, after receiving feedback from the classifier, we measure the SAT loss from Eq. (6) and adapt $\phi$ to minimize this loss. In particular, $\phi$ performs the following update step to facilitate the classifier parameters $\theta$ in the next step:

$$\phi = \phi - \alpha \cdot \frac{1}{B} \sum_{i=1}^{B} \nabla_\phi \mathcal{L}_{\text{cls}} \left( \theta^{(T+1)} \mid x_\theta^{\text{adv}}(i), y(i) \right). \tag{10}$$

Overall, the generator minimizes the loss on the adversarial examples sampled from the perturbation set $S$. Consequently, $\phi$ in Eq. (10) is dependent on $\theta^{(T+1)}$ that depends on $\theta^{(T)}$ (see Eq. (9)), which in turn depends on $x_\phi^{\text{aug}}$ (see Eq. (8)) and acts as a path for the flow of gradients. Similarly, the gradients for $\theta^{(T+1)}$ are chained through the T steps since $\theta^{(T+1)}$ is dependent on $\theta^{(T)}$ that depends on $\theta^{(0)}$, and we use TorchMeta Deleu et al. (2019) for the double backpropagation. We list the detailed algorithm in Algorithm 1. Formally, the overall objective can be summarized as:

$$\min_\phi \ \frac{1}{B} \sum_{i=1}^{B} \mathcal{L}_{\text{cls}} \left( \theta^{(T+1)} \mid x_\theta^{\text{adv}}(i), y(i) \right)$$

$$\text{subject to} \quad \theta^{(T+1)} = \theta^{(T)} - \alpha \cdot \frac{1}{B} \sum_{i=1}^{B} \nabla_\theta \mathcal{L}_{\text{cls}} \left( \theta^{(T)} \mid x_\theta^{\text{adv}}(i), y(i) \right),$$

$$\theta^{(t+1)} = \theta^{(t)} - \alpha \cdot \frac{1}{B} \sum_{i=1}^{B} \nabla_\theta \mathcal{L}_{\text{cls}} \left( \theta^{(t)} \mid x_\phi^{\text{aug}}(i), y(i) \right),$$

$$t = 0, \dots, T-1. \tag{11}$$

To summarize, MNG-AC consists of perturbation sampling to generate adversarial examples. Then, it perturbs the clean examples in a meta-learning framework to explicitly lower the adversarial classification loss on the sampled perturbation. Lastly, the adversarial classifier utilizes the generated samples, adversarial samples and clean samples to optimize the classification and adversarial consistency loss.

**Intuition behind our framework.** Unlike existing adversarial defenses that aim for robustness against single perturbation, our proposed approach targets for a realistic scenario of robustness against multiple perturbations. Our motivation is that meta-learning the noise distribution to minimize the stochastic adversarial classification loss allows to learn the optimal noise to improve multi-perturbation generalization. Based on the assumption that the model should output similar predictions for perturbed versions of the same image, we enforce the adversarial consistency loss, which enforces the label consistency across multiple perturbations. We empirically illustrate that our proposed training scheme increases the smoothness of the model (see Figure 3) and pushes the decision boundary (see Figure 4), which confirms our hypothesis for multi-perturbation generalization.

## 5 EXPERIMENTS

### 5.1 EXPERIMENTAL SETUP

**Baselines and our model.** We compare our method MNG-AC with the standard network (Nat) and state-of-the-art single-perturbation baselines including Madry et al. (2017) (Adv$_\text{p}$) for $\ell_\infty, \ell_1$, and $\ell_2$ norm, Zhang et al. (2019) (TRADES$_\infty$), and Carmon et al. (2019) (RST$_\infty$) for $\ell_\infty$ norm. We also consider state-of-the-art multi-perturbation baselines: namely, we consider Adversarial training with the maximum (see Eq. (3)) (Adv$_\text{max}$), average (Adv$_\text{avg}$) (Tramèr & Boneh, 2019) (see Eq. (4)) over all perturbations, and Multiple steepest descent (MSD) (Maini et al., 2020).

**Datasets.** We evaluate our method on multiple benchmark datasets including CIFAR-10 (Krizhevsky, 2012), SVHN (Netzer et al., 2011) on Wide ResNet 28-10 (Zagoruyko & Komodakis, 2016) and Tiny-ImageNet [2] on ResNet-50 (He et al., 2016) architecture.

**Evaluation setup.** We have evaluated the proposed defense scheme and baselines against perturbations generated by state-of-the-art attack methods. We use the same attack parameters as Tramèr & Boneh (2019) for training and evaluation. We validate the clean accuracy (Acc$_\text{clean}$), the worst (Acc$_\text{adv}^\text{union}$) and average (Acc$_\text{adv}^\text{avg}$) adversarial accuracy across all the perturbation sets for all the models. For $\ell_\infty$ attacks, we use PGD (Madry et al., 2017), Brendel and Bethge (Brendel et al., 2019),

---

[2] https://tiny-imagenet.herokuapp.com/

Table 1: Comparison of robustness against multiple types of perturbations. All the values are measured by computing mean, and standard deviation across three trials upon randomly chosen seeds, the best and second-best results are highlighted in bold and underline respectively. Time denotes the training time in hours. For CIFAR-10 and SVHN, we use $\varepsilon = \{\frac{8}{255}, \frac{2000}{255}, \frac{80}{255}\}$ and $\alpha = \{0.004, 1.0, 0.1\}$ for $\ell_\infty, \ell_1$, and $\ell_2$ attacks respectively. For Tiny-ImageNet, we use $\varepsilon = \{\frac{4}{255}, \frac{2000}{255}, \frac{80}{255}\}$ and $\alpha = \{0.004, 1.0, 0.1\}$ for $\ell_\infty, \ell_1$, and $\ell_2$ attacks respectively. We report the worst-case accuracy for all the attacks and defer the breakdown of all attacks to Appendix B.

| | Model | $\text{Acc}_{\text{clean}}$ | $\ell_\infty$ | $\ell_1$ | $\ell_2$ | $\text{Acc}_{\text{adv}}^{\text{union}}$ | $\text{Acc}_{\text{adv}}^{\text{avg}}$ | Time (h) |
|---|---|---|---|---|---|---|---|---|
| CIFAR-10 | Nat (Zagoruyko & Komodakis, 2016) | 94.7± 0.1 | 0.0± 0.0 | 4.4± 0.8 | 19.4± 1.4 | 0.0± 0.0 | 7.9± 0.3 | **0.4** |
| | Adv$_\infty$ (Madry et al., 2017) | 86.8± 0.1 | 44.9± 0.7 | 12.8± 0.6 | 69.3± 0.4 | 12.9± 0.5 | 42.6± 0.4 | 4.5 |
| | Adv$_1$ | **93.3± 0.4** | 0.0± 0.0 | **78.1± 1.8** | 0.0± 0.0 | 0.0± 0.00 | 25.1± 1.6 | 8.1 |
| | Adv$_2$ | 91.7± 0.2 | 20.7± 0.3 | 27.7± 0.7 | **76.8± 0.4** | 17.9± 0.8 | 47.6± 0.4 | 3.7 |
| | TRADES$_\infty$ (Zhang et al., 2019) | 84.7± 0.3 | 48.9± 0.7 | 17.9± 0.6 | 69.4± 0.3 | 17.2± 0.6 | 45.4± 0.3 | 5.2 |
| | RST$_\infty$ (Carmon et al., 2019) | 88.9± 0.2 | **54.9± 1.8** | 22.0± 0.5 | 73.6± 0.1 | 21.1± 1.0 | 50.2± 0.5 | 58.8 |
| | Adv$_{\text{avg}}$ (Tramèr & Boneh, 2019) | 87.1± 0.2 | 33.8± 0.7 | 49.0± 0.3 | 74.9± 0.4 | 31.0± 1.4 | 52.6± 0.5 | 16.9 |
| | Adv$_{\text{max}}$ (Tramèr & Boneh, 2019) | 85.4± 0.3 | 39.9± 0.9 | 44.6± 0.2 | 73.2± 0.2 | 35.7± 0.3 | 52.5± 0.3 | 16.3 |
| | MSD (Maini et al., 2020) | 82.6± 0.0 | 43.7± 0.2 | 41.6± 0.2 | 70.6± 1.1 | 35.8± 0.1 | 52.0± 0.4 | 16.7 |
| | MNG-AC (Ours) | 81.5± 0.3 | 42.2± 0.9 | 55.0± 1.2 | 71.5± 0.1 | **41.6± 0.8** | **56.2± 0.2** | 11.2 |
| SVHN | Nat (Zagoruyko & Komodakis, 2016) | **96.8± 0.1** | 0.0± 0.0 | 4.4± 0.8 | 19.4± 1.4 | 0.0± 0.0 | 7.9± 0.3 | **0.6** |
| | Adv$_\infty$ (Madry et al., 2017) | 92.8± 0.2 | 46.2± 0.6 | 3.0± 0.3 | 59.2± 0.7 | 3.0± 0.3 | 36.2± 0.3 | 6.2 |
| | Adv$_1$ | 92.4± 0.9 | 0.0± 0.0 | **77.9± 6.3** | 0.0± 0.0 | 0.0± 0.0 | 23.9± 2.1 | 11.8 |
| | Adv$_2$ | 94.9± 0.1 | 18.7± 0.6 | 30.3± 0.3 | **79.3± 0.1** | 16.4± 0.7 | 42.8± 0.2 | 6.1 |
| | TRADES$_\infty$ (Zhang et al., 2019) | 93.9± 0.1 | 49.9± 1.7 | 1.6± 0.3 | 56.0± 1.4 | 1.6± 0.3 | 35.8± 0.6 | 7.9 |
| | RST$_\infty$ (Carmon et al., 2019) | 95.6± 0.0 | **60.9± 2.0** | 0.7± 0.6 | 60.6± 0.6 | 0.7± 0.6 | 40.7± 0.8 | 112.5 |
| | Adv$_{\text{avg}}$ (Tramèr & Boneh, 2019) | 92.6± 0.3 | 17.4± 2.3 | 54.2± 2.9 | 74.7± 0.1 | 16.6± 1.3 | 43.0± 1.0 | 24.1 |
| | Adv$_{\text{max}}$ (Tramèr & Boneh, 2019) | 88.2± 1.3 | 5.9± 1.2 | 48.3± 4.1 | 31.0± 5.0 | 5.8± 1.7 | 26.7± 2.5 | 22.7 |
| | MNG-AC (Ours) | 93.7± 0.1 | 33.7± 1.9 | 47.4± 2.2 | 77.6 ± 1.0 | **30.3± 1.8** | **52.6± 0.5** | 11.9 |
| Tiny-ImageNet | Nat (He et al., 2016) | **62.8± 0.4** | 0.0± 0.0 | 2.7± 0.3 | 12.6± 0.8 | 0.0± 0.0 | 5.1± 0.4 | **0.9** |
| | Adv$_\infty$ (Madry et al., 2017) | 54.2± 0.4 | 29.6± 0.1 | 31.8± 1.0 | 42.5± 0.6 | 19.8± 1.1 | 33.8± 0.1 | 4.3 |
| | Adv$_1$ | 57.8± 0.2 | 10.5± 0.7 | 39.3± 1.0 | 41.9± 0.0 | 10.1± 0.7 | 30.4± 0.1 | 12.9 |
| | Adv$_2$ | 59.5± 0.1 | 5.2± 0.6 | 37.2± 0.4 | **44.9± 0.1** | 5.2± 0.6 | 29.1± 0.0 | 3.7 |
| | TRADES$_\infty$ (Zhang et al., 2019) | 48.2± 0.2 | 28.7± 0.9 | 30.9± 0.2 | 35.8± 0.7 | 26.1± 0.9 | 32.8± 0.1 | 5.8 |
| | Adv$_{\text{avg}}$ (Tramèr & Boneh, 2019) | 56.0± 0.0 | 23.7± 0.2 | 38.0± 0.2 | 44.6± 1.8 | 23.6± 0.3 | 35.4± 0.7 | 26.8 |
| | Adv$_{\text{max}}$ (Tramèr & Boneh, 2019) | 53.5± 0.0 | **29.8± 0.1** | 33.4± 0.3 | 42.4± 1.0 | **29.0± 0.3** | 35.3± 0.4 | 20.8 |
| | MNG-AC (Ours) | 53.1± 0.3 | 27.4± 0.7 | **39.6± 0.7** | 44.8± 0.1 | 27.4± 0.8 | **37.2± 0.6** | 10.4 |

and AutoAttack (Croce & Hein, 2020). For $\ell_2$ attacks, we use CarliniWagner (Carlini & Wagner, 2017), PGD (Madry et al., 2017), Brendel and Bethge (Brendel et al., 2019), and AutoAttack (Croce & Hein, 2020). For $\ell_1$ attacks, we use SLIDE (Tramèr & Boneh, 2019), Salt and pepper (Rauber et al., 2017), and EAD attack (Chen et al., 2018). We provide a detailed description of the experimental setup in Appendix A.

## 5.2 COMPARISON OF ROBUSTNESS AGAINST MULTIPLE PERTURBATIONS

**Results with CIFAR-10 dataset.** Table 1 shows the experimental results for the CIFAR-10 dataset. It is evident from the results that MNG-AC achieves a relative improvement of $\sim 6\%$ and $\sim 4\%$ on the $\text{Acc}_{\text{adv}}^{\text{union}}$ and $\text{Acc}_{\text{adv}}^{\text{avg}}$ metric over the state-of-the-art methods trained on multiple perturbations. Moreover, MNG-AC achieves $\sim 33\%$ reduction in training time compared to the multi-perturbations training baselines. It is also worth mentioning that, MNG-AC also shows an improvement over Adv$_{\text{max}}$, which is fundamentally designed to address the worst perturbation.

**Results with SVHN dataset.** The results for the SVHN dataset are shown in Table 1. We make the following observations from the results: (1) Firstly, MNG-AC significantly outperforms Adv$_{\text{avg}}$, Adv$_{\text{max}}$ by $\sim 14\%$ and $\sim 25\%$ on $\text{Acc}_{\text{adv}}^{\text{union}}$ metric. Furthermore, it achieves an improvement of $\sim 7.2\%$ and $\sim 26\%$ on $\text{Acc}_{\text{adv}}^{\text{avg}}$ metric over Adv$_{\text{avg}}$, Adv$_{\text{max}}$ respectively. (2) Secondly, MNG-AC leads to a $\sim 50\%$ reduction in training time compared to the multi-perturbation training baselines.

Table 2: Ablation study analyzing the significance of SAT, Adversarial Consistency loss (AC) and Meta Noise Generator (MNG). The best results are highlighted in bold.

| | SAT | AC | MNG | $Acc_{clean}$ | $\ell_\infty$ | $\ell_1$ | $\ell_2$ | $Acc_{adv}^{union}$ | $Acc_{adv}^{avg}$ | Time (h) |
|---|---|---|---|---|---|---|---|---|---|---|
| **CIFAR-10** | ✓ | - | - | **87.4**± **0.0** | 34.6± 0.7 | 49.3± 1.0 | **75.5**± **0.1** | 33.9± 0.6 | 53.1± 0.1 | **5.5** |
| | ✓ | ✓ | - | 81.4± 0.0 | 40.4± 0.1 | 53.2± 0.9 | 70.2± 0.1 | 40.1± 0.2 | 54.6± 0.4 | 6.8 |
| | ✓ | ✓ | ✓ | 81.5± 0.3 | **42.2**± **0.9** | **55.0**± **1.2** | 71.5± 0.1 | **41.6**± **0.8** | **56.2**± **0.2** | 11.2 |
| **SVHN** | ✓ | - | - | 92.8± 0.5 | 23.4± 2.4 | 41.3± 4.3 | 71.0± 3.6 | 22.8± 1.5 | 44.9± 1.2 | **7.6** |
| | ✓ | ✓ | - | 92.1± 0.2 | 32.9± 1.8 | 35.4± 1.5 | 77.1± 1.3 | 28.3± 0.1 | 49.6± 0.5 | 9.6 |
| | ✓ | ✓ | ✓ | **93.7**± **0.1** | **35.1**± **1.9** | 47.4± 2.2 | **77.6** ± **1.0** | **30.3**± **1.8** | **52.6**± **0.5** | 11.9 |

Interestingly, MNG-AC achieves significant better performance over $\ell_1$ adversarial training with comparable training time which illustrates the utility of our method over standard adversarial training.

**Results with Tiny-ImageNet.** We also evaluate our method on Tiny-ImageNet to verify that it performs well on complex datasets. In Table 1 we observe that MNG-AC outperforms the multi-perturbation training baselines and achieves comparable performance to the single-perturbation baselines. Only against $\ell_\infty$ perturbations, we notice that $Adv_{max}$ achieves better performance. We believe this is an artefact of the inherent trade-off across multiple perturbations (Tramèr & Boneh, 2019; Schott et al., 2018). Interestingly, MNG-AC even achieves comparable performance to the single perturbation baselines trained on $\ell_1$ and $\ell_2$ norm. This demonstrates the effectiveness of MNG in preventing overfitting over a single attack, and it's generalization ability to diverse types of attacks.

## 5.3 ABLATION STUDIES

**Component analysis.** To further investigate our training scheme, we dissect the effectiveness of various components in Table 2. First, we examine that SAT leads to a $\sim 68\%$ and $\sim 30\%$ reduction in training time over multiple perturbations baselines and MNG-AC for both the datasets, however, it does not improve the adversarial robustness. Then, we analyze the impact of our meta-noise generator by injecting random noise $z \sim \mathcal{N}(0, \mathbf{I})$ to the inputs for the generation of augmented samples. We observe that it significantly improves the performance over the SAT with a marginal increase in the training time. Furthermore, leveraging MNG our combined framework MNG-AC achieves consistent improvements, outperforming all the baselines, demonstrating the efficacy of our meta-learning scheme to defend against multiple perturbations.

**Effect of hyperparameters.** We further analyze the impact of $\beta$ in our augmentation loss (see Eq. (6)) in Figure 2. We evaluate the worst-attack performance across all $\ell_p$ norm adversarial attacks. Our results show that as the value of $\beta$ increases the performance on $\ell_\infty$ and $\ell_1$ attacks improves significantly. In particular, the performance with $\ell_\infty$ and $\ell_1$ attack improve by $4\%$ with an increase in the weight of adversarial consistency loss. However, an increase in $\beta$ also leads to a reduction of $\sim 3\%$ in the robustness against the $\ell_2$ attacks, which is in line with the previous works that have showcased an inherent trade-off between various attacks theoretically and empirically (Tramèr & Boneh, 2019; Schott et al., 2018).

## 5.4 FURTHER ANALYSIS OF OUR DEFENSE

**Results on unforseen adversaries.** We further evaluate our model on various unforeseen perturbations (Kang et al., 2019) namely we evaluate on the Elastic, $\ell_\infty$-JPEG, $\ell_1$-JPEG and $\ell - 2$-JPEG attacks. Note that, even though adversarial training methods do not generalize beyond the threat model, we observe that MNG-VS improves the performance on these adversaries. We compare $MNG_{SAT}$ to the baselines trained with multiple perturbations on the SVHN dataset in Table 3. We notice that even though, $Adv_{max}$ achieves better performance on $\ell_p$-JPEG attacks, it obtains the minimum robustness across the $Acc_{adv}^{union}$ metric. In contrast, MNG-AC generalizes better over both the baselines for the worst-attack in the set of unforeseen perturbations.

**Visualization of loss landscape.** As further qualitative analysis of the effect of MNG-AC, we compare the loss surface of various methods against $\ell_\infty, \ell_1$, and $\ell_2$ norm attack in Figure 3. We can observe that in most of the instances when trained with a single adversary, the adversary can find a direction orthogonal to that explored during training; for example, $\ell_1$ attack results in a non-smooth

Table 3: Performance of MNG-AC against unforseen adversaries on SVHN dataset.

| Model | Elastic | $\ell_\infty$-JPEG | $\ell_1$-JPEG | $\ell_2$-JPEG | $\text{Acc}_{\text{adv}}^{\text{union}}$ |
|---|---|---|---|---|---|
| $\text{Adv}_{\text{avg}}$ | 77.1± 1.1 | 86.1± 1.5 | 78.1± 1.8 | 79.0± 2.0 | 61.5± 1.5 |
| $\text{Adv}_{\text{max}}$ | 60.2± 2.3 | **89.9**± **1.9** | **87.9**± **2.1** | **87.0**± **2.5** | 58.5± 1.5 |
| MNG-AC | **80.8**± **1.0** | 87.7± 1.3 | 76.6± 2.6 | 81.4± 1.2 | **64.3**± **0.5** |

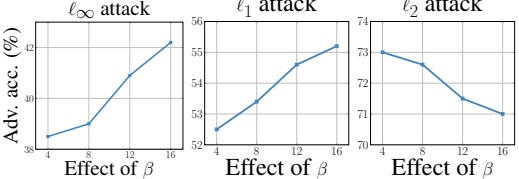

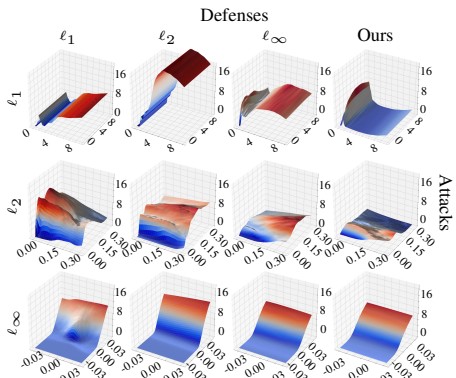

Defenses / Ours / Attacks

Figure 2: Ablation study on the impact of $\mathcal{L}_{\text{aug}}$ on average robustness against $\ell_p$ attacks on CIFAR-10. With an increase in $\beta$, the robustness against $\ell_\infty$ and $\ell_1$ attack increases. However, the robustness of $\ell_2$ decreases showing an inherent trade-off across multiple perturbations.

Figure 3: Visualization of the loss landscapes for the $\ell_1, \ell_2$, and $\ell_\infty$-norm attacks on the SVHN dataset. The rows represent the attacks and columns represent different defenses.

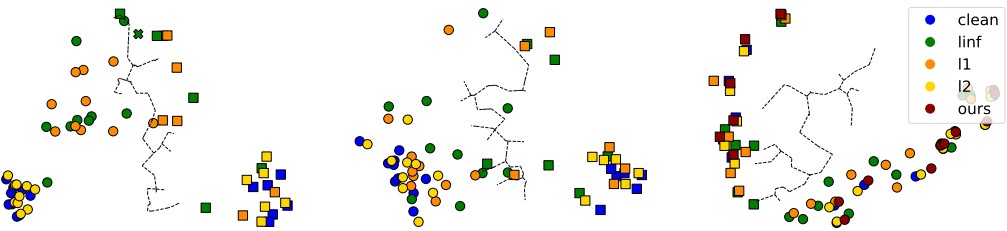

Figure 4: Visualization of decision boundary in the penultimate latent-feature space for $\text{Adv}_{\text{avg}}$ in the left, $\text{Adv}_{\text{max}}$ in the middle, MNG-AC in the right for SVHN dataset on Wide ResNet 28-10 architecture. The two shapes represent different classes in a binary classification task.

loss surface for both $\ell_\infty$ and $\ell_2$ adversarial training. On the contrary, MNG-AC achieves smoother loss surface across all types of attacks which suggests that the gradients modelled by our model are closer to the optimum global landscape. See Appendix B for the loss landscape on CIFAR-10.

**Visualization of decision boundary.** Finally, we visualize the learned decision boundary on binary-classification task across multiple attacks in Figure 4. We can observe that MNG-AC obtains the least error against all the attacks compared to the baselines trained on multiple perturbations. Furthermore, the consistency regularization embeds multiple perturbations onto the same latent space, which pushes them away from the decision boundary that in turn improves the overall robustness. See Appendix B for visualization of the examples generated by our proposed meta-noise generator.

## 6 CONCLUSION

We tackled the problem of robustness against multiple adversarial perturbations. Existing defense methods are tailored to defend against single adversarial perturbation which is an artificial setting to evaluate in real-life scenarios where the adversary will attack the system in any way possible. To this end, we propose a novel *Meta-Noise Generator (MNG)* that learns to stochastically perturb adversarial examples by generating output noise across diverse perturbations. Then we train the model using *Adversarial Consistency loss* that accounts for label consistency across clean, adversarial, and augmented samples. Additionally, to resolve the problem of computation overhead with conventional adversarial training methods for multiple perturbations, we introduce a *Stochastic Adversarial Training (SAT)* which samples a perturbation from the distribution of perturbations. We believe that our method can be a strong guideline when other researchers pursue similar tasks in the future.

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

# A  EXPERIMENTAL SETUP

## A.1  DATASETS

1. **CIFAR-10.** This dataset (Krizhevsky, 2012) contains 60,000 images with 5,000 images for training and 1,000 images for test for each class. Each image is sized $32 \times 32$, we use the Wide ResNet 28-10 architecture (Zagoruyko & Komodakis, 2016) as a base network for this dataset.

2. **SVHN.** This dataset (Netzer et al., 2011) contains 73257 training and 26032 testing images of digits and numbers in natural scene images containing ten-digit classes. Each image is sized $32 \times 32$, we use the Wide ResNet 28-10 architecture similar to the CIFAR-10 dataset as the base network.

3. **Tiny-ImageNet.** This dataset [3] is a subset of ImageNet (Russakovsky et al., 2015) dataset, consisting of 500, 50, and 50 images for training, validation, and test dataset, respectively. This dataset contains $64 \times 64$ size images from 200 classes, we use ResNet50 (He et al., 2016) as a base network for this dataset.

## A.2  TRAINING SETUP

We use the SGD optimizer with momentum 0.9 and weight decay $5 \cdot 10^{-4}$ to train all our models with cyclic learning rate with a maximum learning rate $\lambda$ that increases linearly from 0 to $\lambda$ over first $N/2$ epochs and then decreases linearly from $N/2$ to 0 in the remainder epochs, as recommended by Wong et al. (2020) for fast convergence of adversarial training. We train all the models for 30 epochs on a single machine with four GeForce RTX 2080Ti using WideResNet 28-10 architecture (Zagoruyko & Komodakis, 2016). We use the maximum learning rate of $\lambda = 0.21$ for all our experiments. We use $\beta = 16$ for all the experiments with our meta noise generator. The generator is formulated as a convolutional network with four $3 \times 3$ convolutional layers with LeakyReLU activations and one residual connection from input to output. We use $T = 2$ for all our experiments and all our algorithms are implemented using Pytorch (Paszke et al., 2019) and TorchMeta (Deleu et al., 2019). We use the weight for the KL divergence ($\beta = 6.0$) for TRADES and RST in all our experiments. We replicate all the baselines on SVHN and TinyImageNet since most of the baseline methods have reported their results on MNIST and CIFAR-10. Unfortunately, we found that MSD Maini et al. (2020) did not converge for larger datasets even after our extensive hyperparameter-search. We believe that this is due to the the change in formulation of the inner optimization which leads to a difficulty in convergence for larger datasets. Since the authors also report their results on CIFAR-10, we do not use it as a baseline for other datasets.

## A.3  EVALUATION SETUP

For $\ell_\infty$ perturbations, we use PGD (Madry et al., 2017), Brendel and Bethge attack (Brendel et al., 2019), and AutoAttack (Croce & Hein, 2020). For $\ell_2$ perturbations, we use CarliniWagner attack (Carlini & Wagner, 2017), PGD (Madry et al., 2017), Brendel and Bethge attack (Brendel et al., 2019), and AutoAttack (Croce & Hein, 2020). For $\ell_1$ perturbations, we use SLIDE (Tramèr & Boneh, 2019), Salt and pepper (Rauber et al., 2017), and EAD attack (Chen et al., 2018). For CIFAR-10 and SVHN, we use $\varepsilon = \{\frac{8}{255}, \frac{2000}{255}, \frac{80}{255}\}$ and $\alpha = \{0.004, 1.0, 0.1\}$ for $\ell_\infty, \ell_1$, and $\ell_2$ attacks respectively. For Tiny-ImageNet, we use $\varepsilon = \{\frac{4}{255}, \frac{2000}{255}, \frac{80}{255}\}$ and $\alpha = \{0.004, 1.0, 0.1\}$ for $\ell_\infty, \ell_1$, and $\ell_2$ attacks respectively. We use 10 steps of PGD attack for $\ell_\infty, \ell_2$ during training. For $\ell_1$ adversarial training, we use 20 steps during training and 100 steps during evaluation. We use the code provided by the authors for evaluation against AutoAttack Croce & Hein (2020) and Foolbox (Rauber et al., 2017) library for all the other attacks.

# B  MORE EXPERIMENTAL RESULTS

Due to the length limit of our paper, we provide a breakdown of all the attacks on CIFAR-10 in Table 4, SVHN on Wide ResNet 28-10 in Table 5, Tiny-ImageNet on ResNet50 in Table 6. Besides, we analyze the noise learned by our meta-learning framework on multiple datasets and the loss landscape on the CIFAR-10 dataset.

---

[3] https://tiny-imagenet.herokuapp.com/

Table 4: Summary of adversarial accuracy results for CIFAR-10 on Wide ResNet 28-10 architecture.

| | $\text{Adv}_\infty$ | $\text{Adv}_1$ | $\text{Adv}_2$ | $\text{Trades}_\infty$ | $\text{RST}_\infty$ | $\text{Adv}_{avg}$ | $\text{Adv}_{max}$ | MSD | MNG-AC |
|---|---|---|---|---|---|---|---|---|---|
| Clean Accuracy | 86.8± 0.1 | 93.3± 0.6 | 91.7± 0.2 | 84.7± 0.3 | 88.9± 0.2 | 87.1± 0.2 | 85.4± 0.3 | 82.3± 0.2 | 84.9± 0.3 |
| PGD-$\ell_\infty$ | 46.9± 0.5 | 0.40± 0.7 | 23.6± 0.2 | 52.0± 0.6 | 56.9± 0.1 | 35.2± 0.8 | 42.2± 1.1 | 45.4± 0.4 | 44.5± 1.1 |
| PGD-Foolbox | 54.7± 0.4 | 0.33± 0.6 | 35.3± 0.4 | 57.8± 0.5 | 62.9± 0.3 | 45.0± 0.4 | 50.4± 0.4 | 51.7± 0.8 | 50.8± 0.8 |
| AutoAttack | 44.9± 0.7 | 0.0± 0.0 | 20.7± 0.4 | 48.8± 1.1 | 53.9± 0.3 | 33.8± 0.7 | 39.9± 0.9 | 42.7± 0.2 | 42.8± 0.8 |
| Brendel & Bethge | 49.9± 1.1 | 0.0± 0.0 | 26.8± 0.3 | 52.1± 0.7 | 56.5± 1.8 | 39.6± 0.7 | 45.8± 0.9 | 48.3± 0.4 | 46.8± 0.9 |
| **All $\ell_\infty$ attacks** | 44.9± 0.7 | 0.0± 0.0 | 20.7± 0.3 | 48.9± 0.7 | 54.9± 1.8 | 33.8± 0.7 | 39.9± 0.9 | 43.7± 0.2 | 42.2± 0.9 |
| PGD-$\ell_1$ | 12.8± 0.6 | 91.6± 1.4 | 27.7± 0.7 | 17.9± 0.6 | 22.0± 0.5 | 49.0± 0.3 | 44.6± 0.2 | 46.8± 1.4 | 55.0± 1.2 |
| PGD-Foolbox | 35.2± 0.7 | 92.3± 1.3 | 53.1± 0.5 | 40.3± 0.7 | 44.6± 0.3 | 64.5± 0.2 | 60.7± 0.5 | 60.3± 0.4 | 65.5± 0.1 |
| EAD | 72.9±1.0 | 87.1± 3.3 | 75.9± 1.9 | 80.2± 0.7 | 84.5± 0.2 | 85.7± 0.2 | 83.3± 0.5 | 80.8± 0.1 | 79.3± 0.6 |
| SAPA | 71.5± 0.2 | 80.2± 1.8 | 81.9± 0.5 | 71.4± 0.7 | 76.0± 0.5 | 82.7± 0.1 | 80.0± 0.1 | 76.9± 0.5 | 76.7± 0.4 |
| **All $\ell_1$ attacks** | 12.8± 0.6 | 78.1± 1.8 | 27.7± 0.7 | 17.9± 0.6 | 22.0± 0.5 | 49.0± 0.3 | 44.6± 0.2 | 43.7± 0.2 | 55.0± 1.2 |
| PGD-$\ell_2$ | 78.7± 0.3 | 47.6± 1.6 | 84.6± 0.2 | 77.0± 0.9 | 82.2± 0.2 | 81.5± 0.2 | 79.1± 0.3 | 76.5± 0.1 | 75.6± 0.4 |
| PGD-Foolbox | 74.6± 0.2 | 5.1± 2.1 | 79.8± 0.2 | 73.3± 0.6 | 78.3± 0.2 | 77.6± 0.2 | 75.8± 0.3 | 73.6± 0.5 | 73.4± 0.1 |
| Gaussian Noise | 85.2± 0.4 | 88.5± 1.8 | 90.5± 1.1 | 83.2± 0.3 | 87.8± 0.2 | 86.2± 0.5 | 83.3± 0.3 | 70.9± 1.1 | 79.3± 0.1 |
| AutoAttack | 69.9± 0.4 | 0.0± 0.0 | 76.8± 0.4 | 69.4± 0.3 | 73.7± 0.1 | 74.9± 0.4 | 73.2± 0.2 | 71.9± 0.4 | 71.5± 0.1 |
| Brendel & Bethge | 71.8± 0.9 | 0.0± 0.0 | 78.1± 0.6 | 70.2± 0.1 | 75.0± 0.3 | 75.9± 0.3 | 74.1± 0.4 | 80.4± 0.4 | 72.3± 0.1 |
| CWL2 | 70.5± 0.2 | 0.1± 0.0 | 77.2± 0.5 | 69.7± 0.3 | 74.2± 0.1 | 74.6± 1.2 | 73.5± 0.2 | 71.1± 1.1 | 71.0± 0.1 |
| **All $\ell_2$ attacks** | 69.3± 0.4 | 0.0± 0.0 | 76.8± 0.4 | 69.4± 0.3 | 73.6± 0.1 | 74.9± 0.4 | 73.2± 0.2 | 70.6± 1.1 | 71.5± 0.1 |
| $\text{Acc}_{adv}^{union}$ | 12.9± 0.5 | 0.0± 0.0 | 17.9± 0.8 | 17.2± 0.6 | 21.1± 1.0 | 31.0± 1.4 | 35.7± 0.3 | 35.8± 0.1 | **41.6± 0.8** |
| $\text{Acc}_{adv}^{avg}$ | 42.6± 0.4 | 25.1± 1.6 | 47.6± 0.4 | 45.4± 0.3 | 50.2± 0.5 | 52.6± 0.5 | 52.5± 0.3 | 52.0± 0.4 | **56.2± 0.2** |

**Visualization of learned noise.** To demonstrate the learning ability of our meta-noise generator, we visualize the learned noise by our generator during training. We present representative samples projected on various $\ell_p$ norms and datasets in Figure 5 where each sample is projected to their respected norm-ball $\mathcal{B}(x, \varepsilon)$ around $x$ with radius $\varepsilon$. From the figure, we can observe that our meta-noise generator incorporates the features by different attacks and learns diverse input-dependent noise distributions across multiple adversarial perturbations by explicitly minimizing the adversarial loss across multiple perturbations during meta-training. Overall, it combines two approaches that are complementary to each other and leads to a novel input-dependent learner for generalization across diverse attacks.

**Visuaization of loss landscape on CIFAR-10.** Figure 6 shows the visualization of loss landscape of various methods against $\ell_\infty, \ell_1$, and $\ell_2$ norm attack for CIFAR-10 dataset on Wide ResNet 28-10 architecture. We vary the input along a linear space defined by the norm of the gradient where x and y-axes represent the perturbation added in each direction, and the z-axis represents the loss. Similar to the SVHN dataset, we can observe that the loss is highly curved for multiple perturbations in the vicinity of the data point $x$ for the adversarial training trained with a single perturbation, which reflects that the gradient poorly models the global landscape. In contrast, MNG-AC achieves smoother loss surface across all types of $\ell_p$ norm attacks.

Table 5: Summary of adversarial accuracy results for SVHN dataset on Wide ResNet 28-10 architecture.

| | $\text{Adv}_\infty$ | $\text{Adv}_1$ | $\text{Adv}_2$ | $\text{Trades}_\infty$ | $\text{RST}_\infty$ | $\text{Adv}_{\text{avg}}$ | $\text{Adv}_{\text{max}}$ | MNG-AC |
|---|---|---|---|---|---|---|---|---|
| Clean Accuracy | 92.8± 0.1 | 92.4± 1.6 | 94.9± 0.0 | 93.9± 0.0 | 95.6± 0.0 | 92.6± 0.1 | 88.2± 1.6 | 93.4± 0.0 |
| PGD-$\ell_\infty$ | 49.1± 0.1 | 3.2± 2.4 | 29.4± 0.1 | 55.5± 1.4 | 66.9± 0.8 | 22.4± 3.1 | 36.6± 2.0 | 40.5± 0.1 |
| PGD-Foolbox | 60.7± 0.4 | 2.5± 1.9 | 47.6± 0.6 | 66.4± 1.1 | 73.8± 0.3 | 32.5± 3.2 | 49.9± 0.0 | 57.5± 1.8 |
| AutoAttack | 46.2± 0.6 | 0.0± 0.0 | 18.9± 0.5 | 49.9± 1.8 | 61.0± 2.0 | 17.6± 2.6 | 17.5± 0.9 | 33.7± 1.8 |
| Brendel & Bethge | 51.6± 0.7 | 0.0± 0.0 | 22.9± 0.8 | 55.8± 1.5 | 65.6± 1.2 | 20.2± 2.9 | 6.3± 2.3 | 40.0± 0.3 |
| **All $\ell_\infty$ attacks** | 46.2± 0.6 | 0.0± 0.0 | 18.7± 0.6 | 49.9± 1.7 | 60.9± 2.0 | 17.4± 2.3 | 5.9± 1.2 | 33.7± 1.9 |
| PGD-$\ell_1$ | 3.1± 0.3 | 95.0± 1.8 | 30.5± 0.4 | 1.7± 0.3 | 0.7± 0.6 | 55.8± 2.1 | 48.4± 2.9 | 44.5± 3.2 |
| PGD-Foolbox | 19.9± 0.8 | 94.6± 0.4 | 57.5± 0.1 | 15.5± 0.2 | 11.3± 0.5 | 79.2± 3.4 | 85.4± 3.2 | 75.2± 2.8 |
| EAD | 65.7± 2.1 | 87.8± 1.9 | 82.3± 1.2 | 51.5± 2.9 | 60.4± 0.8 | 84.8± 2.4 | 84.5± 3.8 | 86.2± 2.2 |
| SAPA | 79.4± 0.8 | 77.3± 5.2 | 87.3± 0.1 | 73.5± 1.0 | 86.2± 0.5 | 88.5± 0.6 | 80.9± 4.0 | 89.9± 1.6 |
| **All $\ell_1$ attacks** | 3.0± 0.3 | 77.9± 6.3 | 30.3± 0.3 | 1.6± 0.3 | 0.7± 0.6 | 54.2± 2.9 | 48.3± 4.1 | 47.4± 2.2 |
| PGD-$\ell_2$ | 81.6± 0.5 | 3.9± 1.4 | 87.8± 0.2 | 83.9± 0.8 | 85.3± 0.2 | 85.6± 0.6 | 84.3± 1.1 | 90.4± 0.6 |
| PGD-Foolbox | 73.2± 0.2 | 1.9± 1.8 | 82.8± 0.6 | 75.0± 0.7 | 76.0± 0.3 | 80.6± 0.1 | 60.1± 0.8 | 86.1± 0.1 |
| Gaussian Noise | 92.1± 0.2 | 16.5± 4.2 | 94.2± 0.2 | 93.3± 1.4 | 94.2± 0.6 | 92.2± 0.2 | 83.8± 0.6 | 93.2± 0.4 |
| AutoAttack | 59.0± 0.7 | 0.0± 0.0 | 79.3± 0.1 | 56.4± 1.3 | 60.7± 0.6 | 75.6± 0.1 | 40.0± 2.3 | 78.0± 0.8 |
| Brendel & Bethge | 68.2± 0.5 | 0.0± 0.0 | 81.0± 0.1 | 64.8± 0.9 | 68.1± 0.5 | 76.4± 0.4 | 32.7± 3.8 | 78.4± 0.4 |
| CWL2 | 63.5± 0.8 | 0.1± 0.1 | 80.1± 1.4 | 61.4± 0.3 | 63.9± 0.2 | 76.8± 0.1 | 55.3± 5.2 | 80.9± 0.9 |
| **All $\ell_2$ attacks** | 59.2± 0.7 | 0.0± 0.0 | 79.3± 0.1 | 56.0± 1.4 | 60.6± 0.6 | 74.7± 0.1 | 31.0± 5.0 | 77.6± 1.0 |
| $\text{Acc}_{\text{adv}}^{\text{union}}$ | 3.0± 0.3 | 0.0± 0.0 | 16.4± 0.7 | 1.6± 0.3 | 0.7± 0.6 | 16.6± 1.3 | 5.8± 1.7 | **30.3± 1.8** |
| $\text{Acc}_{\text{adv}}^{\text{avg}}$ | 36.2± 0.3 | 23.9± 2.1 | 42.8± 0.2 | 35.8± 0.6 | 40.7± 0.8 | 43.0± 1.0 | 26.7± 2.5 | **52.6± 0.5** |

Table 6: Summary of adversarial accuracy results for Tiny-ImageNet on ResNet50 architecture.

| | $Adv_\infty$ | $Adv_1$ | $Adv_2$ | $Trades_\infty$ | $Adv_{avg}$ | $Adv_{max}$ | MNG-AC |
|---|---|---|---|---|---|---|---|
| Clean Accuracy | 54.2± 0.1 | 57.8± 0.2 | 59.8± 0.1 | 48.2± 0.2 | 56.0± 0.2 | 53.5± 0.0 | 53.1± 0.1 |
| PGD-$\ell_\infty$ | 32.1± 0.0 | 11.5± 1.2 | 17.9± 1.1 | 32.2± 0.4 | 25.0± 0.6 | 32.0± 0.6 | 29.3± 0.3 |
| PGD-Foolbox | 34.6± 0.4 | 17.2± 0.1 | 5.2± 0.6 | 34.1± 0.2 | 34.0± 0.2 | 28.3± 0.1 | 32.3± 0.3 |
| AutoAttack | 29.6± 0.1 | 10.1± 0.7 | 16.3± 0.3 | 28.7± 0.9 | 23.7± 0.2 | 30.0± 0.1 | 27.7± 0.4 |
| Brendel & Bethge | 32.7± 0.1 | 14.6± 0.8 | 20.8± 0.6 | 31.0± 0.9 | 28.1± 0.2 | 33.2± 0.5 | 31.5± 0.6 |
| **All $\ell_\infty$ attacks** | 29.6± 0.1 | 10.5± 0.7 | 5.2± 0.6 | 28.7± 0.9 | 23.7± 0.2 | 29.8± 0.1 | 27.4± 0.7 |
| PGD-$\ell_1$ | 32.0± 1.1 | 39.3± 0.9 | 37.2± 0.2 | 31.1± 0.3 | 38.0± 0.1 | 33.6± 0.4 | 39.0± 0.9 |
| PGD-Foolbox | 40.0± 0.8 | 44.8± 0.2 | 45.2± 0.2 | 37.6± 0.9 | 44.7± 1.5 | 40.6± 0.1 | 45.0± 0.2 |
| EAD | 52.3± 1.5 | 56.3± 0.6 | 57.3± 0.0 | 46.7± 0.9 | 54.6± 0.9 | 51.2± 0.2 | 52.7± 0.3 |
| SAPA | 46.5± 0.9 | 52.9± 0.7 | 53.5± 1.2 | 40.8± 0.1 | 50.3± 1.1 | 46.6± 0.1 | 49.3± 0.4 |
| **All $\ell_1$ attacks** | 31.8± 1.0 | 39.3± 1.0 | 37.2± 0.4 | 30.9± 0.2 | 38.0± 0.2 | 33.4± 0.3 | 39.6± 0.7 |
| PGD-$\ell_2$ | 48.5± 1.1 | 49.1± 0.1 | 51.8± 1.8 | 42.6± 0.7 | 49.9± 1.7 | 47.0± 0.3 | 49.1± 0.4 |
| PGD-Foolbox | 45.6± 0.4 | 45.2± 0.4 | 47.7± 0.7 | 41.0± 0.3 | 47.0± 1.3 | 44.9± 0.4 | 47.0± 0.2 |
| Gaussian Noise | 52.5± 1.3 | 56.1± 0.6 | 57.6± 0.3 | 46.4± 0.9 | 54.4± 0.8 | 51.1± 0.0 | 52.1± 0.5 |
| AutoAttack | 42.4± 0.8 | 41.9± 0.0 | 44.6± 0.6 | 38.9± 0.8 | 44.4± 1.3 | 42.4± 0.9 | 44.6± 0.4 |
| Brendel & Bethge | 43.7± 0.4 | 44.4± 0.1 | 46.6± 1.1 | 39.2± 0.7 | 45.1± 1.6 | 43.6± 0.4 | 45.4± 0.1 |
| CWL2 | 43.5± 1.3 | 44.8± 1.1 | 47.5± 0.7 | 39.5± 0.4 | 46.8± 1.9 | 43.4± 0.1 | 46.0± 0.4 |
| **All $\ell_2$ attacks** | 42.5± 0.6 | 41.9± 0.0 | 44.9± 0.1 | 35.8± 0.7 | 44.6± 0.1 | 42.4± 1.0 | 44.8± 0.1 |
| $Acc_{adv}^{union}$ | 19.8± 1.1 | 10.1± 0.7 | 5.2± 0.6 | 26.1± 0.9 | 23.6± 0.3 | **29.0± 0.3** | 27.4± 0.8 |
| $Acc_{adv}^{avg}$ | 33.8± 0.1 | 30.4± 0.1 | 29.1± 0.0 | 32.8± 0.1 | 35.4± 0.7 | 35.3± 0.4 | **37.2± 0.6** |

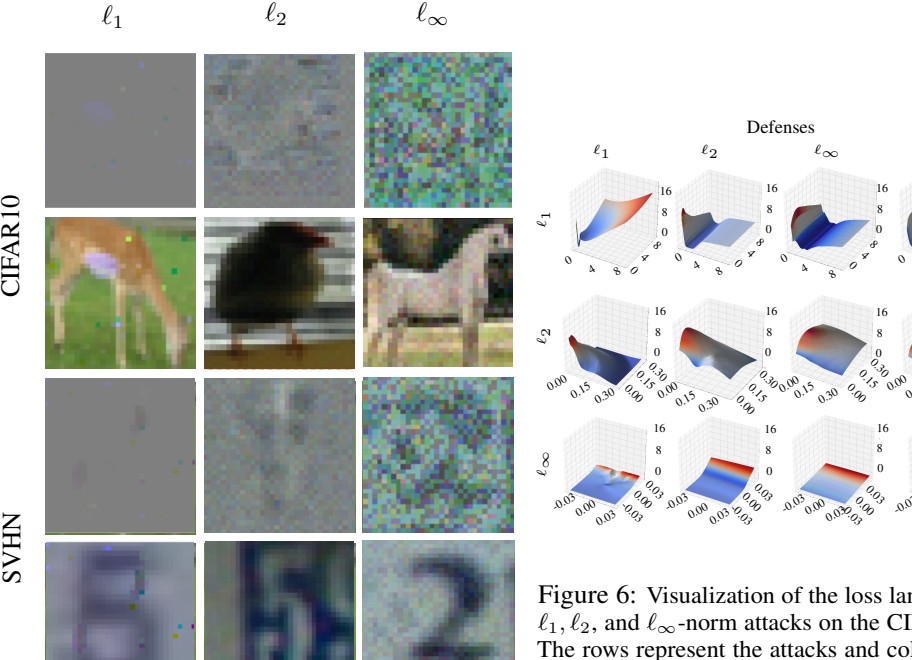

Figure 6: Visualization of the loss landscapes for the $\ell_1, \ell_2$, and $\ell_\infty$-norm attacks on the CIFAR-10 dataset. The rows represent the attacks and columns represent different defenses. We can observe that that MNG-AC obtains smooth loss surface across all $\ell_p$-norm attacks.

Figure 5: Visualization of the generated noise by MNG along with the perturbed samples on $\ell_1, \ell_2$, and $\ell_\infty$-norm attacks for CIFAR-10 and SVHN dataset.

