# OpenReview forum: "Learning to Generate Noise for Multi-Attack Robustness"
_ICLR.cc/2021/Conference — Reject_

### Official Review · AnonReviewer3 · 2020-10-27

**Rating:** 6
**Confidence:** 1

**Review:**

1. Summary

The authors propose a new method to improve robustness to adversarial examples under various norms (L1, L2 and LInf). Their method combines adversarial training with an adversarial noise generator. They improve upon adversarial training in a multi norm setting by choosing one norm at random for each sample, instead of computing an adversarial for all norms, thus significantly reducing the training time. They additionally improve robustness by regularizing model features between the standard image, the adversarially perturbed image and a perturbation of the image created with an adversarial noise generator.


2. Strengths
+ The method is based on adversarial training. As far as I know and as the authors note this is the only method that reliably leads to more robust models.
+ The authors attack their models with  a range of attacks that to the best of my knowledge are state-of-the art.
+ The method apparently works in the multi norm setting.

3. Weaknesses
- I was missing an intuitive description why the adversarial noise should improve robustness to adversarial attacks. I was only aware of it as a method to improve corruption robustness.
- I was not always sure if I got everything correctly in sections 4 and 5.3. I think I got it but I sometimes missed a figure. It may e.g. be helpful to include the losses in Figure 1 or make a separate figure. Especially why the MNG was trained the way it is was a bit unclear for me.


4. Recommendation

I think this paper is an accept but as I don't work with adversarial examples I am not at all confident in that assessment. From the discussions with people who work on adversarial examples new defenses are usually broken very quickly and there is a number of papers which break numerous defenses. The method is however based on adversarial training which to my knowledge is the only robust method so far and the used attacks seem valid. So I am definitely leaning towards accept but the opinion of a real expert would be highly appreciated as I feel not at all qualified to assess the validity of papers on adversarial examples.


5. Questions/Recommendations
- Is there a difference between the M(eta)NG and the A(dversarial)NG from Rusak et. al. 2020?


6. Additional feedback
- None as the paper is pretty well written.

---

> ### Author Response · Authors · 2020-11-11
> **Response to R3**
>
> We sincerely appreciate your constructive comments. We respond to your main concerns below:
>
> 1. I was missing an intuitive description of why the adversarial noise should improve robustness to adversarial attacks. I was only aware of it as a method to improve corruption robustness.
>
> - It is important to note that simply adding noise **would not improve the robustness to adversarial attacks** (see our comparison with A(dversarial)NG below), MNG **explicitly learns an optimal noise distribution**to prevent overfitting and to promote the generalization across multiple perturbations. Additionally, adversarial noise acts as a noise regularization technique, which is a common technique to improve the generalization in deep neural networks. Further, we have added a separate paragraph to highlight the illustration of our training scheme in Section 4 of the revision of our paper.
> ---
> 2. Is there a difference between the M(eta)NG and the A(dversarial)NG from Rusak et al. 2020?
>
> - It is important to note that A(dversarial)NG (Rusak et al. 20) learns the noise projected on $\ell_2$ norm-ball to confuse the classifier, in contrast maximally, we **meta-learn the noise distribution to compliment the generalization across multiple $\ell_p$ perturbations.** Further, we show that **A(dversarial)NG fails to defend against multiple adversarial perturbations**below, which demonstrates the efficiency of M(eta)NG over A(dversarial)NG:
>
> | Dataset       	| Model  	| Acc$_{\rm clean}$ 	| $\ell_\infty$ 	|  $\ell_1$ 	|  $\ell_2$ 	|
> |---------------	|--------	|:-----------------:	|:-------------:	|:---------:	|:---------:	|
> | CIFAR-10      	| MNG-AC 	|     81.5+-0.3     	|   42.2+-0.9   	| 55.0+-1.2 	| 71.5+-0.1 	|
> | CIFAR-10      	| ANG    	|     94.6+-0.0     	|   0.1+-0.00   	|  0.1+-0.0 	|  2.9+-0.9 	|
> | SVHN          	| MNG-AC 	|     93.7+-0.1     	|   35.1+-1.9   	| 47.4+-2.2 	| 77.6+-1.0 	|
> | SVHN          	| ANG    	|     96.8+-0.1     	|    0.2+-0.0   	|  7.3+-0.5 	| 33.9+-1.5 	|
> | Tiny-ImageNet 	| MNG-AC 	|     53.1+-0.3     	|   27.4+-0.7   	| 39.6+-0.7 	| 44.8+-0.1 	|
> | Tiny-ImageNet 	| ANG    	|     62.8+-0.3     	|    0.2+-0.1   	|  3.4+-0.4 	| 13.4+-0.6 	|
>
> ---
> 3. Why the MNG was trained the way it is was a bit unclear for me.
> - Our objective was to learn optimal noise distribution that could explicitly minimize the loss of multiple adversarial perturbations and promote label consistency across multiple perturbations. A standard approach was to use a bilevel optimization to train the adversarial classifier with MNG. However, bilevel optimization for adversarial training was **computationally costly.** As a result, we adopted an alternative scheme where we first update the model parameters on the augmented samples for $T$ steps, to explicitly increase the influence of the augmented samples. Then we perform a one-step lookahead to model the adaptaion of the adversarial classifier in the presence of augmented examples. Lastly, after receiving the feedback from the classifier, we update $\phi$ to explicitly minimize the adversarial loss to promote the adversarial robustness of the classifier in the next step.
>
> 4. I don't work with adversarial examples I am not at all confident in that assessment. From the discussions with people who work on adversarial examples new defenses are usually broken very quickly and there is a number of papers which break numerous defenses. The method is however based on adversarial training which to my knowledge is the only robust method so far and the used attacks seem valid. So I am definitely leaning towards accept but the opinion of a real expert would be highly appreciated as I feel not at all qualified to assess the validity of papers on adversarial examples.
>
> - We understand your concern, and we would like to highlight that we have evaluated our proposed method on all the state-of-the-art attacks that exist in the literature. We believe that our evaluation can be a firm guideline when other researchers pursue the evaluation of defenses that are robust against multiple perturbations in the future. Further, as you rightly mentioned our defense is based on adversarial training, which is the only robust method that has withstood the stronger set of attacks.

---

> > ### Author Response · Authors · 2020-11-23
> > **Response to R3 (2)**
> >
> > Dear Reviewer 3,
> >
> > Could you please go over our responses and the revision and let us know if there is any other information we should provide since we can have interactions with you only by this Tuesday (24th)? We have responded to your comments and faithfully reflected them in the revision. We sincerely thank you for your time and efforts in reviewing our paper, and your insightful and constructive comments.
> >
> > Thanks, Authors

---

### Official Review · AnonReviewer1 · 2020-10-28
**Thorough Multi-attack Robustness Evaluation and Clever Adversarial Training**

**Rating:** 6
**Confidence:** 5

**Review:**

This paper addresses a timely issue in adversarial robustness - efficient training of robust models against multiple adversarial perturbations. The authors propose a combination of three techniques: stochastic adversarial training (SAT), meta noise generator (MNG), and adversarial consistency (AC) loss for efficient training, and evaluate the robustness using multiple L1, L2, and Linf norm-bounded attacks and three datasets (CIFAR-10, SVHN, and Tiny Imagenet). The results show improved multi-attack robustness over several baselines (including single-attack and multiple-attack models) and reduced training time. Ablation studies are also performed to illustrate the utility of each component of the proposed model. Overall, this paper provides very detailed evaluations involving multiple datasets, attacks, baselines, and robustness metrics. I find the results convincing and important, and also find sufficient novelty in the proposed training method.

The strengths (S) and weaknesses (W) of this submission are summarized below.

S1. The proposal of MNG and AC is effective and novel.
S2. The evaluation is thorough and convincing.
S3. The proposal improves both robustness and training efficiency in most cases.

W1. The adversarial consistency (AC) loss is never defined explicitly. Based on equation (5), it is hard to understand how AC "represents the Jensen-Shannon Divergence (JSD) among the posterior distributions" when considering three distributions, P_clean, P_adv, and P_aug. More clarification is needed.

W2. Although the results show improved multi-attack robustness, it will be great if the authors can add more intuition on why the proposed training method leads to performance improvement. Based on the ablation study,  it seems that the role of SAT and MNG is to reduce overfitting in robustness to encourage generalization, rather than optimization over the worst-case scenarios.

W3. The considered multi-attack setting is still limited to different Lp norm perturbation constraints. Although the authors showed improved robustness over unforeseen attacks, the authors should also discuss how the proposed method can generalize to different attacks beyond Lp norms.

---

> ### Author Response · Authors · 2020-11-11
> **Response to R1**
>
> We sincerely appreciate your constructive comments. We respond to your main concerns below:
>
> 1) The adversarial consistency (AC) loss is never defined explicitly.
> - We apologize for the confusion. We have updated the revision with the explicit definition of the Adversarial Consistency (AC) loss in Equation 6.
> ---
> 2) Although the results show improved multi-attack robustness, it will be great if the authors can add more intuition on why the proposed training method leads to performance improvement. Based on the ablation study, it seems that the role of SAT and MNG is to reduce overfitting in robustness to encourage generalization, rather than optimization over the worst-case scenarios.
>
> - As you mentioned, SAT and MNG indeed play a critical role to reduce overfitting in robustness to encourage generalization. Intuitively, MNG acts as a **noise regularization technique,** and SAT promotes generalization across multiple perturbations **due to its stochasticity.** Further, we have added a separate paragraph to highlight the illustration of our training scheme in Section 4 of the revision of our paper.
>
> - Additionally, we would like to clarify that unlike the max strategy, MNG and SAT **do not optimize over the worst-case scenarios.** MNG learns an **input-dependent optimal noise distribution to lower adversarial error across all the perturbations**that does not necessarily correspond to any of the attack perturbations.
> ---
> 3) The considered multi-attack setting is still limited to different Lp norm perturbation constraints. Although the authors showed improved robustness over unforeseen attacks, the authors should also discuss how the proposed method can generalize to different attacks beyond Lp norms.
>
> - We agree that the evaluation of attacks beyond $\ell_p$ norms is interesting, and we would like to point out that the unforeseen adversaries consist of Elastic attack and JPEG attacks which **do not belong to the standard family of $\ell_p$ attacks.**

---

> > ### Comment · AnonReviewer1 · 2020-11-17
> > **My review concerns are well addressed**
> >
> > I thank the authors for providing the clarification and for revising the submission to address my review comments. I have no additional questions.
> >
> > One minor follow-up discussion: In the response, you wrote "MNG and SAT do not optimize over the worst-case scenarios." I believe you mean "MNG does not optimize over the worst-case scenarios". My understanding is that SAT is nothing but a stochastic version of full-batch worst-case optimization.

---

> > > ### Author Response · Authors · 2020-11-18
> > > **Response to R1 (2)**
> > >
> > > Thank you so much for your quick and timely response to our rebuttal.
> > >
> > > 1. In the response, you wrote "MNG and SAT do not optimize over the worst-case scenarios." I believe you mean "MNG does not optimize over the worst-case scenarios". My understanding is that SAT is nothing but a stochastic version of full-batch worst-case optimization.
> > > - We sincerely apologize for the confusion. We want to clarify that "worst-case optimization" in multi-perturbation training implies **optimizing over the strongest perturbation**from the perturbation set, that is, optimizing over the attack that leads to the maximum loss. However, it requires the computation of all the attacks in the perturbation set, which significantly increases the computational cost. In contrast, as you pointed out, the SAT is a stochastic version of the full-batch worst-case optimization, which **prevents overfitting**on a single perturbation set by stochastically sampling an attack from a given perturbation set and achieves comparable performance to multi-perturbation training with **significant lower computation cost.**
> > > - Further, MNG meta-learns the noise distribution to **minimize the adversarial loss on the sampled attack**and not over the worst-case scenarios, as worst-case optimization does not necessarily lead to an optimal solution (see Adv$_{max}$ results on SVHN dataset in Table 1).

---

### Official Review · AnonReviewer2 · 2020-10-28
**Promising results, but method is not clear**

**Rating:** 5
**Confidence:** 4

**Review:**

Summary
=======
The authors propose a number of techniques to learn models which are adversarially robust to multiple perturbations. These involve a noise generator, a loss to enforce consistency, as well as a stochastic variant of adversarial training. With these changes, they are able to produce improvements to robust accuracy to multiple perturbation types.


Overall, I get the idea and the empirical results seem promising. However, the structure and writing of the paper is at times rather confusing, and there are a lot of missing details. If the code were not supplied, it would be difficult in the current state to reproduce the method from the paper. Perhaps due to this, the specifics of the key component, the meta noise generator, are still rather opaque to me. Perhaps the authors can clarify, and I am happy to follow up afterwards.

Comments for discussion
=======================
The majority of my confusion lies in section 4, for the specifics of the meta noise generator and parts of the algorithm in general. I am otherwise well acquainted with the relevant literature.

1) Augmented examples (x_aug) are generated by adding noise from the MNG and projecting it onto some ball B. It is not clear to me what ball this is since the authors are considering multiple perturbations. Is it a random type? Or a joint projection? I assume it is at least one of the perturbations being considered, or is that incorrect?

2) Similarly, in the algorithm, the authors generate adversarial examples (x_adv) by sampling a random attack. I could not find what set of attacks were being sampled from, or what the sampling distribution is (I checked the appendix as well).

3) The generator is apparently updated to minimize the classifier loss on the adversarial examples as written in Equation (8). However, the adversarial examples are generated from some unspecified set of attacks, which implies that the set of attacks actually depends on the generator somehow. Is this supposed to be the classifier loss on the augmented samples? If not, then how do the adversarial examples depend on the generator?

4) The consistency loss involves clean, adversarial, and augmented posterior distributions. There are no details on these distributions: are these simply the softmax of the logits? Or is a generative model that outputs a distribution being used?

5) On a more fundamental level, what is the motivation behind training the generator to minimize the classifier loss? Why would we want to do this over random sampling? What's to prevent a degenerate solution of simply learning to produce a zero perturbation (and thus always producing clean examples, which can achieve low loss)?


Minor comments
==============
I have checked the supplementary material and the authors have included the code for running their experiments. Ideally, this would also include pre-trained model weights.

Update
======
After much effort, I can say that I understand the paper. The edits appear to have incorporated all the identified missing information. I have thus updated my confidence and slightly improved my score, however I am not confident that the current presentation of the approach will be understandable by a reader without contacting the authors, given that the difficulty I had in understanding the paper (and my initial confidence) stemmed primarily from missing information and poor presentation for the approach. Although the results do seem to improve upon past work, its impact will suffer if it is difficult to understand for a non-reviewer reader. I would be more confident if a fresh set of eyes could understand the details of the work without having to go to the authors to clarify so many details.

---

> ### Author Response · Authors · 2020-11-11
> **Response to R2**
>
> We sincerely appreciate your constructive comments. We respond to your main concerns below:
>
> 1. Augmented examples (x_aug) are generated by adding noise from the MNG and projecting it onto some ball B. It is not clear to me what ball this is since the authors are considering multiple perturbations. Is it a random type? Or a joint projection? I assume it is at least one of the perturbations being considered, or is that incorrect?
>
> - $\mathcal{B}(x, \varepsilon)$ refers to the **norm-ball of the specific attack**sampled by Stochastic Adversarial Training (SAT). That is, if the sampled attack is an $\ell_2$ attack, then $\mathcal{B}$ denotes the $\ell_2$ norm ball, and if the sampled attack is an $\ell_1$ attack, then $\mathcal{B}$ is the $\ell_1$ norm ball. As MNG learns the noise to minimize the adversarial loss, it is essential to project the generated noise on the same norm-ball. We have clarified this point in the revision.
> ---
> 2. Similarly, in the algorithm, the authors generate adversarial examples (x_adv) by sampling a random attack. I could not find what set of attacks were being sampled from, or what the sampling distribution is (I checked the appendix as well).
>
> - We apologize for the confusion. In this work, the sampling distribution corresponds to the **$\ell_p$-bounded perturbations.** Still, it is important to note that unlike the average and max strategy, MNG + SAT can be applied to any distribution of attacks with a constant cost. We have clarified this point in the revision.
> ---
> 3. The generator is apparently updated to minimize the classifier loss on the adversarial examples as written in Equation (8). However, the adversarial examples are generated from some unspecified set of attacks, which implies that the set of attacks actually depends on the generator somehow. Is this supposed to be the classifier loss on the augmented samples? If not, then how do the adversarial examples depend on the generator?
>
> - This is a critical misunderstanding. **Adversarial examples do not depend on the generator;** instead, the one-step update in Eq. (8) is essential to do a lookahead for adapting the model parameters in the presence of the noise-augmented samples. Note that augmented samples are different from the adversarial examples (please refer Figure 1) and our contribution is to optimally generate augmented examples to improve the robustness against multiple perturbations explicitly.
> ---
> 4. The consistency loss involves clean, adversarial, and augmented posterior distributions. There are no details on these distributions: are these simply the softmax of the logits? Or is a generative model that outputs a distribution being used?
>
> - As you mentioned, the distributions are the softmax of the logits of the clean, adversarial and augmented samples where the augmented samples are the output of the Meta Noise Generator (MNG). We have incorporated this point in the revision.
> ---
> 5. What is the motivation behind training the generator to minimize the classifier loss? Why would we want to do this over random sampling? What's to prevent a degenerate solution of simply learning to produce a zero perturbation (and thus always producing clean examples, which can achieve low loss)?
>
> - Firstly, we would like to clarify that we **do not train the generator to minimize the classifier loss;** instead, the generator learns an optimal noise distribution in a meta-learning training scheme to **minimize the adversarial classification loss**where adversarial classification loss is the loss on the sampled attack from the distribution of attacks.
>
> - Secondly, the motivation behind training the generator to minimize this objective is to explicitly learn the noise distribution essential for generalization across multiple perturbations, that might not necessarily correspond to any of the attack perturbations. Furthermore, our algorithm to improve the generalization across multiple perturbations is also motivated by the popular phenomenon of noise regularization being a common technique to improve the generalization performance of deep neural networks. In contrast, even though random sampling helps in generalization, it leads to a suboptimal solution (see Table 2).
>
> - Lastly, our meta-learning training scheme prevents the degenerate solution, as producing clean examples would not result in a lower loss on multiple adversarial perturbations.
> ---
> 6. I have checked the supplementary material, and the authors have included the code for running their experiments. Ideally, this would also include pre-trained model weights.
>
> - Thank you for pointing this out. Due to the size limit of the supplementary material, we could not provide the pre-trained models. We provide the pre-trained model weights here: https://drive.google.com/file/d/1kVfOZ2CrhSzgzlS6gK4AntNZhIUosvfz/view?usp=sharing

---

> > ### Comment · AnonReviewer2 · 2020-11-16
> > **Suggestions and unanswered questions**
> >
> > Thanks for the response! Based on the response, I have a couple suggestions and some outstanding questions that weren't answered in the response, the latter of which I bring up again below.
> >
> > Re 1) Thank you for clarifying this. I re-read the revision and it still wasn't clear in the text (if there was a part which made this explicit please point it to me as I may have missed it). Since there is extra space, I would recommend adding these details to the Algorithm box as the existing references to Equation 4 in line 3 is ambiguous. Not only does this make the norm ball unclear, but also Eq. 4 is actually the full robust optimization problem and not only the adversarial examples generation problem.
> >
> > Re 2) I understand that the distribution is picking from Lp bounded perturbations, which tells me that the support of the distribution is the set of Lp balls. However, I still have no idea what the actual distribution is from the response, and still could not find this information in the revision.
> >
> > Re 3+5) There seems to have been a mistake here in the equation I was referring to. I am referring to the update step for the generator, which is now equation (9) in the revision. I understand at this point that the intention of the authors in the response was to convey that the adversarial examples and the generator do not depend on each other. But Equation (9) is minimizing the *generator* parameters with respect to the classification loss evaluated on the *adversarial examples*, which seems to be in direct contradiction to the claim "we do not train the generator to minimize the classifier loss". Since the adversarial examples do not depend on the generator, I don't see how there can be a gradient with respect to the generator parameters \phi here, since the classification loss of the adversarial examples has nothing to do with the generator.
> >
> > Since this is such a critical misunderstanding, some further revision here is likely needed (i.e. perhaps this has something to do with the total loss, which is curiously not used at all in any of the steps of the algorithm).

---

> > > ### Author Response · Authors · 2020-11-17
> > > **Response to R2 (2)**
> > >
> > > We thank you for your clarifying questions and suggestions.
> > >
> > > Re 1) I re-read the revision and it still wasn't clear in the text. I would recommend adding these details to the Algorithm box as the existing references to Equation 4 in line 3 is ambiguous. Not only does this make the norm ball unclear, but also Eq. 4 is actually the full robust optimization problem and not only the adversarial examples generation problem.
> > >
> > > - We sincerely apologize for the confusion. We have highlighted all our changes in blue, in the revision. According to your suggestions, **we have updated Algorithm 1**. In particular, we have **explicitly defined the attack generation**in Eq. (1) and we refer to this equation for the generation of adversarial examples in Algorithm 1 using the sampled attack. We have also updated the notation of norm-ball $\mathcal{B}(x,\varepsilon)$ to $\mathcal{B}_{\mathcal{A}}(x,\varepsilon)$ to represent the norm ball for a specific attack type $\mathcal{A}$.
> > > ---
> > >
> > > Re 2) I understand that the distribution is picking from Lp bounded perturbations, which tells me that the support of the distribution is the set of Lp balls. However, I still have no idea what the actual distribution is from the response, and still could not find this information in the revision.
> > >
> > > - We apologize for the confusion. We have updated the notations and **explicitly defined a perturbation set**$S$ and the **attack sampling procedure**in Eq. (5) of the revision. We have further clarified and stated this in the Algorithm 1 of the revision as well.
> > >
> > > ---
> > >
> > > Re 3.1)  I understand at this point that the intention of the authors in the response was to convey that the adversarial examples and the generator do not depend on each other.
> > >
> > > - We apologize for the misunderstanding. We want to clarify that **our intention was not to convey that**adversarial examples and the generator does not depend on each other. The adversarial examples do not depend on the generator, as the adversarial examples are simply generated using the PGD attack (see Eq. (1) in the revision), but the generator does depend on the adversarial examples as it generates a sample to minimize adversarial classification loss across multiple perturbations (see Eq. (10) in the revision), via meta-learning.
> > >
> > > - Please note that while the meta-generator generates a sample that minimizes the adversarial classification loss across multiple perturbations, it is **not necessarily an adversarial example**. The generated sample simply needs to be effective in minimizing the adversarial classification loss and enforcing label consistency across samples from multiple attacks, and clean samples. Consequently, it pushes the decision boundary (see Figure 4) and enforces a smooth and robust network across multiple perturbations.
> > >
> > > ---
> > >
> > > Re 3.2) Equation (9) is minimizing the generator parameters with respect to the classification loss evaluated on the adversarial examples, which seems to be in direct contradiction to the claim "we do not train the generator to minimize the classifier loss".
> > >
> > > - Please note that the classifier loss or the **classification loss denotes the loss on clean examples $(\mathcal{L}_{cls}(\theta \mid x^{clean},y))$** and the adversarial classifier loss or the **adversarial classification loss denotes the loss on adversarial examples generated by our sampling procedure $(\mathcal{L}_{cls}(\theta \mid x^{adv},y))$.**
> > >
> > > - Thus, our claim that "we do not train the generator to minimize the classifier loss" implies that **we do not train the generator to minimize the loss on clean examples.** Instead, the generator **explicitly learns an optimal noise distribution to minimize the loss across multiple adversarial perturbations** (as denoted by Eq. (9) in previous revision or Eq. (10) in the current revision and our previous comment).
> > >
> > > ---
> > >
> > > Re 3.3) Since the adversarial examples do not depend on the generator, I don't see how there can be a gradient with respect to the generator parameters \phi here, since the classification loss of the adversarial examples has nothing to do with the generator.
> > >
> > > - As mentioned in Re 3.1), the adversarial examples do not depend on the generator as the adversarial examples are generated using the PGD attack. Still, the generator  depends on the adversarial examples sampled from the perturbation set $S$.  Consequently, $\phi$ in Eq. (10) is dependent on $\theta^{(T+1)}$ that depends on $\theta^{(T)}$ (see Eq. (9)) which in turn depends on $x^{\rm{aug}}$ (see Eq. (8)) and acts as a path for the flow of gradients.
> > >
> > > ---
> > >
> > > Re 3.4) Since this is such a critical misunderstanding, some further revision here is likely needed (i.e. perhaps this has something to do with the total loss, which is curiously not used at all in any of the steps of the algorithm).
> > >
> > > - The step 6 of the Algorithm 1 in the revision (step 5 in the previous version of the paper) updates $\theta$ to minimize the total loss. We have explicitly mentioned this in the main text.

---

> > > > ### Comment · AnonReviewer2 · 2020-11-19
> > > > **Thanks for the answers; but how far back do gradients go for all steps?**
> > > >
> > > > I thank the authors for clarifying my questions. I now have a better understanding of the proposed approach, though this brings up a few additional questions which are not quite clear yet.
> > > >
> > > > I understand now that the gradient with respect to \phi of equation 9 is propagated throughout all the previous steps to x_aug. I would recommend that the authors either make this explicit or update their notation, as the current set of equations as written do not show any dependence on \phi beyond the implicit dependence through x_aug and so this is ambiguous. There is a \phi in the classifier loss of Equation 11, but this is not ideal and somewhat confusing as it's really a direct dependency of x_aug.
> > > >
> > > > Due to this ambiguity, it is also unclear what terms are being backpropagated through in each equation. Since \phi is being backpropagated all the way back to x_aug, what about the other steps for \theta? Are the update steps for \theta also chaining gradients all the way back to \theta_0? Are you using a double backpropagation library here to do this? If these are just single step gradients taken locally with respect to the current iteration of the parameter, then the notation here is inconsistent with \phi and needs to be more carefully presented.

---

> > > > > ### Author Response · Authors · 2020-11-19
> > > > > **Response to R2 (3)**
> > > > >
> > > > > We sincerely thank you for responsiveness and timely response to our rebuttal.
> > > > > > I understand now that the gradient with respect to \phi of equation 9 is propagated throughout all the previous steps to x_aug. I would recommend that the authors either make this explicit or update their notation, as the current set of equations as written do not show any dependence on \phi beyond the implicit dependence through x_aug and so this is ambiguous. There is a \phi in the classifier loss of Equation 11, but this is not ideal and somewhat confusing as it's really a direct dependency of x_aug.
> > > > > - We thank you for your suggestions. We have now **explicitly specified the flow of gradients**in the revision (see the paragraph below Eq. (10) in the revision). Further, to explicitly show the dependence of $\phi$ on $x^{aug}$, we have updated our notation from $x^{aug}$ to $x^{aug}\_{\phi}$.
> > > > > - Additionally, we have also updated the notation of $x^{adv}$ to $x^{adv}_{\theta}$ to explicitly show that adversarial examples $x^{adv}$ are generated using the PGD attack on the classifier with **parameters $\theta$**and are independent of the generator.
> > > > > ---
> > > > > > Due to this ambiguity, it is also unclear what terms are being backpropagated through in each equation. Since \phi is being backpropagated all the way back to x_aug, what about the other steps for \theta? Are the update steps for \theta also chaining gradients all the way back to \theta_0? Are you using a double backpropagation library here to do this? If these are just single step gradients taken locally with respect to the current iteration of the parameter, then the notation here is inconsistent with \phi and needs to be more carefully presented.
> > > > > - We sincerely apologize for the confusion. As you rightly pointed out, the update steps for $\theta$ **chain back to $\theta^{(0)}$**, since $\theta^{(T+1)}$ is dependent on $\theta^{(T)}$ that depends on $\theta^{(0)}$, and we use TorchMeta [1] for the double backpropagation.
> > > > >
> > > > > We hope that we have clarified your concerns. Please let us know if you have any more concerns or would like us to elaborate on any of the above points. We are happy to provide more clarifications if there is anything still unclear.
> > > > >
> > > > > [1] Deleu et al., 2019 Torchmeta: A Meta-Learning library for PyTorch (https://arxiv.org/pdf/1909.06576.pdf)

---

> ### Author Response · Authors · 2020-11-25
> **Summary of Response for R2**
>
> We thank you for your efforts in reviewing our paper, as well as for the insightful and constructive comments. Since the discussion phase will end in a few hours, we provide a summary of our response to your review below:
>
> - We updated our notation for the norm-ball $B$ to $B_{\mathcal{A}}$. Further, we explicitly defined a perturbation set $S$ and the attack sampling procedure in Eq. (5) of the revision.
> - We elaborated Algorithm 1, incorporating your suggestions in the revision.
> - We added a separate paragraph to highlight the intuition of our proposed framework in the revision.
> - We clarified the flow of gradients for $\phi$ and $\theta$ in the revision.
> - We clarified the meta-learning objective for the generator during the rebuttal period.
> - Finally, we also provided the pre-trained weights to promote reproducibility.
>
> We hope that we have satisfactorily addressed all your comments and suggestions, both in the responses and in the revision. We thank you again for your comments, which helped us significantly improve the quality and clarity of the paper.
>
> Best regards,
> Authors

---

### Official Review · AnonReviewer4 · 2020-10-29
**Initial review**

**Rating:** 6
**Confidence:** 5

**Review:**

In this paper, the authors propose a novel meta-learning framework that explicitly learns to generate noise to improve model robustness (against multiple types of attacks). The results indicate that the proposed approach improves on the state-of-the-art.

Overall, the paper is well written. However some details are missing and this could make the paper hard to reproduce. The experiments could be expanded.

1) There is a significant amount of work about using generative models to build adversarial examples. The literature review only focuses on classical adversarial robustness and robustness against multiple adversaries. I'd recommend making a review of these approaches, even if they are orthogonal to the one proposed in this paper (e.g., [1,2,3])
2) In Eq. (6), what is \mathcal{B}(x, \epsilon). Since there is multiple threat models, I am assuming that it is selected at random between l_1, l_2 and l_inf (like SAT).
3) The number of inner steps T seems to be critical (as it will trade-off gradient precision with compute). However, I don't see any study on this in the paper. Also, it is not clear which value was used for the experiments.
4) Looking at Eq. (7), it seems like backpropagation through the T inner steps is necessary to compute the gradients w.r.t. \phi. This seems overly expensive and I find surprising that adv_avg and adv_max take so much longer to train.
5) Concerning Eq. (7), as a curiousity, have authors considered implicit differentiation [4] ?
6) The experiments are run using 30 epochs which is rather on slim side. E.g., RST_inf should reach about 59% robust accuracy with 200 epochs of training (with 30 epochs it only reaches 55%). I'm curious as to whether the comparison with the proposed approach is unfair (e.g., Adv_inf sees a single adv example per batch, whereas MNG-AC sees 2).
7) It's not entirely clear to me why beta negatively affects l_2 robustness. I'd assume that if the model was only trained against l_2, then there might be an optimal value for beta that is different that the one from Fig. 2. In general, it would be interesting to see on MNG-AC does if different subsets of threats are used.
8) The l_2 loss landscapes seem more noisy that what they should be. Also it's unclear why the axes are centered for l_inf and not for l_2 (explain how these are generated).
9) In Table 5, MNG-AC achieves 35.1% against all l_inf attacks, but only 33.7% against AutoAttack. Am I missing something?

Details:

A) It would helpful to the reader to have the epsilon values written on top of the different tables. The captions could be expanded to include more details.
B) Visuaization -> Visualization

[1] https://openreview.net/pdf?id=SJeQEp4YDH: GAT: Generative Adversarial Training for Adversarial Example Detection and Robust Classification
[2] https://arxiv.org/pdf/1801.02610: Generating Adversarial Examples with Adversarial Networks
[3] https://arxiv.org/pdf/1710.10766: PixelDefend: Leveraging Generative Models to Understand and Defend against Adversarial Examples
[4] https://arxiv.org/pdf/1911.02590: Optimizing Millions of Hyperparameters by Implicit Differentiation

---

> ### Author Response · Authors · 2020-11-11
> **Response to R4**
>
> We sincerely appreciate your constructive comments. We respond to your main concerns below:
>
> 1. I'd recommend making a review of generative models to build adversarial examples, even if they are orthogonal to the one proposed in this paper (e.g., [1,2,3])
>
> - Thank you for the helpful suggestion. We have provided a detailed review of generative models for adversarial robustness in the revision.
> ---
> 2. In Eq. (6), what is $\mathcal{B}(x, \epsilon)$. Since there are multiple threat models, I am assuming that it is selected at random between l_1, l_2 and l_inf (like SAT).
>
> - As you rightly mentioned, $\mathcal{B}(x, \varepsilon)$ refers to a **random norm-ball (like SAT)**. That is, if the sampled attack is an $\ell_2$ attack, then $\mathcal{B}$ denotes the $\ell_2$ norm ball, and if the sampled attack is an $\ell_1$ attack, then $\mathcal{B}$ is the $\ell_1$ norm ball. Still, it is important to note that  $\mathcal{B}(x, \varepsilon)$ is the norm-ball of the attack sampled by Stochastic Adversarial Training (SAT), as MNG learns the noise to minimize the adversarial loss, it is essential to project the generated noise on the same norm-ball. We have clarified this in the revision.
> ---
> 3. The number of inner steps T seems to be critical. However, I don't see any study on this in the paper. It is not clear which value was used for the experiments.
>
> - We used $T=2$ for all our experiments to keep the training cost minimum. We empirically found that larger values of T do not provide a significant increase in the robustness while leading to a significant increase in the training cost. We provide a comparison with different values of $T$ below:
> | Model   	| $\ell_\infty$ 	| $\ell_1$  	| $\ell_2$  	| Time (h) 	|
> |---------	|---------------	|-----------	|-----------	|----------	|
> | $T = 1$ 	| 41.5+-0.8     	| 55.1+-0.9 	| 71.8+-0.2 	| 9.4      	|
> | $T = 2$ 	| 42.2+-0.9     	| 55.0+-1.2 	| 71.5+-0.1 	| 11.2     	|
> | $T = 4$ 	| 42.4+-0.8     	| 55.6+-1.1 	| 71.0+-0.2 	| 14.6     	|
> | $T = 8$ 	| 42.6+-1.0     	| 55.3+-1.2 	| 71.0+-0.1 	| 18.9     	|
>
> ---
> 4. The experiments are run using 30 epochs which is rather on slim side. E.g., RST_inf should reach about 59% robust accuracy with 200 epochs of training (with 30 epochs it only reaches 55%). I'm curious as to whether the comparison with the proposed approach is unfair (e.g., Adv_inf sees a single adv example per batch, whereas MNG-AC sees 2).
>
> - It is essential to note that RST uses ~5 million data points for CIFAR-10 and SVHN, and it took us 4 days with four GeForce RTX 2080Ti to train with 30 epochs. Since it takes **more than 24 days**to finish RST with 200 epochs, we did not evaluate it. Furthermore, we would like to clarify that MNG-AC does not see 2 examples per batch, the lookahead in Equation 8. occurs with a meta-model, and the classifier update occurs only once. (Please see Line 393 in train_MNG.py in our code for more details).
> ---
> 5. It's not entirely clear to me why beta negatively affects l_2 robustness. In general, it would be interesting to see what MNG-AC does if different subsets of threats are used.
>
> - We would like to clarify that it is not a general statement that beta negatively affects $\ell_2$ robustness, instead $\beta$ controls the trade-off between multiple perturbations. We will do our best to get the results for different subsets of threats by the end of the rebuttal deadline.
> ---
> 6. The l_2 loss landscapes seem more noisy that what they should be. Also it's unclear why the axes are centred for l_inf and not for l_2 (explain how these are generated).
>
> - We apologize for the confusion. The axes are centred for all the $\ell_p$ norms; we will further clean the plots in the revision. To generate these plots, we vary the input along a linear space defined by the $\ell_p$ norm of the gradient where x and y-axes represent the perturbation added in each direction, and the z-axis represents the loss.
> ---
> 7. In Table 5, MNG-AC achieves 35.1% against all l_inf attacks, but only 33.7% against AutoAttack. Am I missing something?
>
> - Thank you for pointing this out. We have fixed this discrepancy in the revision.
> ---
> 8. Concerning Eq. (7), as a curiosity, have authors considered implicit differentiation [4]?
>
> - Thank you for suggesting relevant work. However, due to the **high computational cost of hypergradients,** we did not evaluate this direction of work. We will add a reference to this work in the final draft, and a comparison with it should be an interesting problem for future work.
> ---
> 9. The caption could be expanded to include epsilon values. B) Visuaization -> Visualization
> - Thank you for the suggestions, we have updated the caption and typo in the revision.
>
> Additionally, to promote reproducibility of our work, we provide the pre-trained model weights here: https://drive.google.com/file/d/1kVfOZ2CrhSzgzlS6gK4AntNZhIUosvfz/view?usp=sharing

---

> > ### Comment · AnonReviewer4 · 2020-11-20
> > **Thank you**
> >
> > Thank you for your answers. Here are a few more points related to the new manuscript:
> >
> > 1. Eq (1) in the revised manuscript is not the typical PGD procedure used by Madry et al. In particular, it is unclear how the argmax is solved here. It would also be good to define $\mathcal{B}$ directly here too.
> >
> > 2. The SAT explanation remains unclear. My suggestion would be to introduce the symbol $\mathcal{A}_\mathcal{B}$ that performs PGD using $\mathcal{B}$. The authors can then use $\mathcal{B}_1$, ... The current definitions seems cyclic, e.g. $\mathcal{A}_\mathcal{B}$ is used in line 5 of Alg. 1 and $\mathcal{B}_\mathcal{A}$ is used elsewhere. It seems to be simpler to sample the norm-balls rather than the attacks notation-wise.
> >
> > 3. It is not clear that gradients are stopped from $\theta_{T - 1}$. Is that correct?
> >
> > 4. Would it make sense to sample the different attacks with different rate (rather than uniformly)?
> >
> > 5. Tiny-ImaegeNet -> Tiny-ImageNet
> >
> > 6. I realize from the caption of Table 1 that l-inf uses eps=0.03, which is much smaller than the usual 8/255 that other work use. I'm really intrigued as to why RST/TRADES perform so poorly. It also seems that l-inf and l-2 perturbations are compatible and only l-1 seems poor when using RST/TRADES. Would it be possible to train a TRADES or RST model that uses l-inf for half of the examples and l-1 for the other half?

---

> > > ### Author Response · Authors · 2020-11-21
> > > **Response to R4 (3)**
> > >
> > > We sincerely appreciate your feedback. We respond to your concerns below:
> > >
> > > 1. Eq (1) in the revised manuscript is not the typical PGD procedure used by Madry et al. In particular, it is unclear how the argmax is solved here.
> > > - Please note that the typical PGD procedure used by Madry et al. is limited to $\ell_\infty$ norm where it uses the sign of the gradient. In contrast, we have used the standard notation for projected steepest descent, which is common in many papers in the literature (for, eg. see Eq. (1) in Wong et al. [2], Eq. (3) in Maini et al. [2]).
> > > ---
> > > 2. The current definitions seems cyclic, e.g. $\mathcal{A}_\mathcal{B}$ is used in line 5 of Alg. 1 and $\mathcal{B}_\mathcal{A}$ is used elsewhere. It seems to be simpler to sample the norm-balls rather than the attacks notation-wise.
> > > - We thank you for your suggestion. However, there exists a counter-example for this corresponding notation, for instance, we can sample a norm-ball $\mathcal{B}=\ell_1$, and then project a sampled attack $\ell_2$ on $\mathcal{B}=\ell_1$. In contrast, we sample an attack $\mathcal{A}$ and associate it’s corresponding norm-ball to avoid this confusion. We would also like to point out that we use $\mathcal{B}_\mathcal{A}$ in line 5 of Alg. 1 for consistency of our notation.
> > > - Additionally, the perturbation set $S$ could generally also consist of non $\ell_p$ perturbations where it is required to only sample an attack $\mathcal{A}$ and the norm-ball $\mathcal{B}$ is not required.
> > > ---
> > > 3. It is not clear that gradients are stopped from \theta_{T-1}. Is that correct
> > > - We hope that we clarified this point in our previous response (2).
> > > ---
> > > 4. Would it make sense to sample the different attacks with different rate (rather than uniformly)?
> > > -Thank you for raising this. We show the results with different perturbation sets on CIFAR-10 with WideResNet 28-10 as you suggested before to illustrate this effect. It can be observed the trained model is comparatively less robust on the attack, which was not used in training. We believe that the comparable performance on $\ell_2$ might be due to the similar attack strength across different attacks, and would have a different outcome with a higher epsilon.
> > >
> > > | Model                      | $\ell_\infty$     |   $\ell_1$      |   $\ell_2$      |
> > > |------------------------    |:-------------:    |:-----------:    |:-----------:    |
> > > | $\ell_\infty + \ell_1$     |  42.0 +- 0.2      |  56.3 +-0.7     | 70.3 +- 0.1     |
> > > | $\ell_\infty + \ell_2$     |  45.5 +- 0.6      | 31.8 +- 0.8     | 72.3 +- 0.2     |
> > > | $\ell_1 + \ell_2$          |  26.7 +- 0.4      | 57.3 +- 0.7     | 72.6 +- 0.1     |
> > > ---
> > > 5.  I realize from the caption of Table 1 that l-inf uses eps=0.03, which is much smaller than the usual 8/255 that other work use. I'm really intrigued as to why RST/TRADES perform so poorly. It also seems that l-inf and l-2 perturbations are compatible and only l-1 seems poor when using RST/TRADES. Would it be possible to train a TRADES or RST model that uses l-inf for half of the examples and l-1 for the other half?
> > > - First, we want to clarify that 8/255 = 0.031, which is the same as eps=0.03, and we use the same epsilon for adversarial training and evaluation. This can also be verified from the `attack_pgd` function in our attached supplementary code. Second, it is not always true that only $\ell_1$ seems poor when using RST/TRADES (please see the results on SVHN and Tiny-ImageNet in Table 1). We are currently training the TRADES  model that uses l-inf for half of the examples and l-1 and we will try our best to add the results before the rebuttal deadline.
> > >
> > > We hope that we have clarified your concerns. Please let us know if you have any more concerns or would like us to elaborate on any of the above points. We are happy to provide more clarifications if there is anything still unclear.
> > >
> > > [1] Wong et al., Wasserstein Adversarial Examples via Projected Sinkhorn Iterations, ICML 19
> > >
> > > [2] Maini et al., Adversarial Robustness Against the Union of Multiple Perturbation Models, ICML 20

---

> > > > ### Comment · AnonReviewer4 · 2020-11-23
> > > > **Thank you for the clarifications**
> > > >
> > > > > we have used the standard notation for projected steepest descent
> > > >
> > > > One of the issues with the equation here is that $\mathcal{B}$ is not clearly defined and thus we do not know which norm to use for $||v||$. Again, this is a relatively minor comment, but maybe the norm should be specified as $||v||_p$.
> > > >
> > > > > there exists a counter-example for this corresponding notation, for instance, we can sample a norm-ball $\mathcal{B} = \ell_1$, and then project a sampled attack $\ell_2$ on $\mathcal{B} = \ell_1$.
> > > >
> > > > I thought the paper only used one specific attack for each $p$-norm where $p = \\{\infty, 1, 2\\}$. I understand that the authors want to keep the notation general, but it is usually harder to follow (especially when it is not needed in the rest of the paper).
> > > >
> > > > > we show the results with different perturbation
> > > >
> > > > This is very interesting. Thank you for taking the time to try this.
> > > >
> > > > > we want to clarify that 8/255 = 0.031, which is the same as eps=0.03
> > > >
> > > > $8/255 = 0.03137\ldots \neq 0.031 \neq 0.03$. These are subtle difference that make comparison with other work more difficult. Please, make sure that the text/captions are clear that these are the perturbation radii used (anyone skimming through the paper will assume 8/255).
> > > >
> > > > > We are currently training the TRADES model that uses l-inf for half of the examples and l-1
> > > >
> > > > Ultimately, these are just suggestions that are triggered by my own curiosity. Do not worry about having such results before the end of the rebuttal period.

---

> > > > > ### Author Response · Authors · 2020-11-23
> > > > > **Response to R4 (4)**
> > > > >
> > > > > We sincerely appreciate your comments and timely response.
> > > > > > The chaining of gradients was not very clear from the manuscript. Maybe adding a small footnote could help the reader.
> > > > > - Thank you for your suggestion, we have explicitly specified this in the paragraph above the overall objective in the revision.
> > > > >
> > > > > > One of the issues with the equation here is that $\mathcal{B}$ is not clearly defined and thus we do not know which norm to use for $||v||$ . Again, this is a relatively minor comment, but maybe the norm should be specified as $||v||_p$ .
> > > > > - We apologize for this confusion. We have defined and updated the notation of $B$ to $\mathcal{B}\_{\mathcal{A}}$ and $||v||$ to $||v||_{\mathcal{A}}$ in Eq. (1) in the revision.
> > > > >
> > > > > > These are subtle difference that make comparison with other work more difficult.
> > > > > - We are sincerely sorry for the confusion. We have updated the text/caption accordingly in our revision (we use $8/255$ for $\ell_\infty$ for all our training models and evaluations.)
> > > > >
> > > > > > Additional results.
> > > > > - According to your previous suggestions, we experimented with TRADES for ~10 hours of training time (70 epochs instead of 30 used in our current revision). We provide the results in our response. We did not observe any significant differences in our results compared to when trained with 70 epochs. We apologize that we could not do multiple runs for 70 epochs due to the shortage of compute and time.
> > > > > | Dataset  	| Model              	| $\ell_\infty$ 	|   $\ell_1$  	|   $\ell_2$  	|
> > > > > |----------	|--------------------	|:-------------:	|:-----------:	|:-----------:	|
> > > > > | CIFAR-10 	| TRADES (30 epochs) 	|  48.9 += 0.7  	| 17.9 +- 0.6 	| 69.4 +- 0.3 	|
> > > > > | CIFAR-10 	| TRADES (70 epochs) 	|      48.9     	|     15.6    	|      69.7       	|
> > > > > | SVHN     	| TRADES (30 epochs) 	|  49.9 +- 1.7  	|  1.6 +- 0.3 	| 56.0 +- 1.4 	|
> > > > > | SVHN     	| TRADES (70 epochs) 	|      47.1     	|     3.9     	|     52.2    	|

---

> > ### Comment · AnonReviewer4 · 2020-11-20
> > **Direct answers**
> >
> > Thank you for the additional results and updates to the manuscript. Here are some direct answers related to the authors' response.
> >
> > > We empirically found that larger values of T do not provide a significant increase in the robustness while leading to a significant increase in the training cost.
> >
> > I'm assuming that this could be because the gradients are not propagated through the T steps, but rather just the last one. Have the authors tried back-propagating through more steps?
> >
> > > It is essential to note that RST uses ~5 million data points for CIFAR-10 [...] it takes more than 24 days to finish RST with 200 epochs.
> >
> > I'm not quite sure if my understanding of RST (from Carmon et al.) matches the authors. I believe they used 200 CIFAR-10 equivalent epochs. Hence the models sees 50,000 * 200 images throughout training (the training batch is split 50% between supervised and unsupervised datapoints). I understand however that running 200 epochs can be prohibitively expensive, but - for fairness - I would also be interested in seeing Adv_inf, TRADES or RST trained for ~10 hours.
> >
> > In the table, it is stated that RST took 50+ hours. Hence it does seem like the authors ran it for about 300 CIFAR-10 equivalent epochs. Yet, robust accuracy is only 55% (instead of the 59% obtained in the original publication).
> >
> > > The axes are centred for all the $\ell_p$  norms.
> >
> > I still don't see how this is the case, the l-1 and l-2 row seem un-centered, while l-inf is centered.
> >
> > > due to the high computational cost of hypergradients
> >
> > Please, do consider trying it as it is much less computationally expensive that one might expect.

---

> > > ### Author Response · Authors · 2020-11-21
> > > **Response to R4 (2)**
> > >
> > > We sincerely thank you for responsiveness and timely response to our rebuttal.
> > >
> > > 1.  I'm assuming that this could be because the gradients are not propagated through the T steps, but rather just the last one. Have the authors tried back-propagating through more steps?
> > > - We would like to clarify that the gradients are chained through the T steps since $\theta^{(T+1)}$ is dependent on $\theta^{(T)}$ that depends on $\theta^{(0)}$, and we use TorchMeta [1] for the double backpropagation.
> > > ---
> > > 2. I'm not quite sure if my understanding of RST (from Carmon et al.) matches the authors. I believe they used 200 CIFAR-10 equivalent epochs. Hence the models sees 50,000 * 200 images throughout training (the training batch is split 50% between supervised and unsupervised datapoints). I understand however that running 200 epochs can be prohibitively expensive, but - for fairness - I would also be interested in seeing Adv_inf, TRADES or RST trained for ~10 hours.
> > > In the table, it is stated that RST took 50+ hours. Hence it does seem like the authors ran it for about 300 CIFAR-10 equivalent epochs. Yet, robust accuracy is only 55% (instead of the 59% obtained in the original publication).
> > >
> > > - First, we want to clarify the notation that RST (Carmon et al.) sees (50k + 500k) * 200 images during training where 50k images are the standard supervised points and 500k unsupervised Tiny Images. Each epoch of RST takes around 3 hours of computation, so we don’t have the compute available to run it for 200 epochs, and 10 hours of RST is not sufficient for convergence. We are currently training TRADES for 3x training time and will update our response before the rebuttal deadline.
> > > - Second, please note that RST does not generalize to multiple perturbations and thus is not a competitor but instead can be combined with our method. We trained MNG-AC + RST after our submission we provide the results for CIFAR-10 in our response below. We can observe that with the same set of hyper-parameters and training steps (30 epochs), MNG-AC + RST significantly improves the performance on both $\ell_1$ and $\ell_2$ attack. We will add these results in our final revision, and we believe that this should further strengthen our paper as it is not feasible to combine RST with the current multi-perturbation training methods due to the significant training cost.
> > >
> > > | Model        	        | $\ell_\infty$ 	|   $\ell_1$  	|   $\ell_2$  	|
> > > |--------------	        |:-------------:	|:-----------:	|:-----------:	|
> > > | RST          	        |  54.9 +- 1.8  	| 22.0 +- 0.5 	| 73.6 +- 0.1 	|
> > > | MNG-AC       	|  42.2 +- 0.9  	| 55.0 +- 1.2 	| 71.5 +- 0.1 	|
> > > | MNG-AC + RST 	|  46.2 +- 1.2  	| 62.6 +- 1.4 	| 80.9 +- 0.1 	|
> > > ---
> > > 3. I still don't see how this is the case, the l-1 and l-2 row seem un-centered, while l-inf is centered.
> > > - We are incredibly sorry for this confusion. This is due to the discrepancy in the labels of the graphs, and we will fix this in the update of our paper.
> > > ---
> > > 4.  Please, do consider trying it as it is much less computationally expensive that one might expect.
> > > - Please note that the exact meta-learning algorithm itself is not our main contribution; we can also use a bilevel-optimization similar to the MAML framework [2]. However, due to the high computation cost of bilevel-optimization, we adopted an online approximation [3, 4] for computational efficiency. To incorporate your suggestions, we will do our best to implement hyper-gradients, but we are not sure whether it will be possible to include it by the rebuttal deadline.
> > >
> > > [1] Deleu et al., 2019 Torchmeta: A Meta-Learning library for PyTorch (https://arxiv.org/pdf/1909.06576.pdf)
> > >
> > > [2] FInn et al., 2017, Model-Agnostic Meta-Learning for Fast Adaptation of Deep Networks, ICML 2017
> > >
> > > [3] Jang et al., 2019 Learning What and Where to Transfer, ICML 2019
> > >
> > > [4] Shu et al., 2019 Meta-Weight-Net: Learning an Explicit Mapping For Sample Weighting, NeurIPS 2019

---

> > > > ### Comment · AnonReviewer4 · 2020-11-23
> > > > **Thank you for the quick reply**
> > > >
> > > > > the gradients are chained through the T steps
> > > >
> > > > This was not very clear from the manuscript. Maybe adding a small footnote could help the reader. I also see that I was not the only reviewer confused by this. I am happy to hear that the gradients are propagated through the whole chain.
> > > >
> > > > > we want to clarify the notation that RST (Carmon et al.) sees (50k + 500k) * 200 images during training where 50k images are the standard supervised points and 500k unsupervised Tiny Images.
> > > >
> > > > This is really as minor issue, but I do not think the authors' understanding is correct (the supplementary material in Carmon et al. indicates otherwise: "composing every batch from equal parts labeled and unlabeled data"). To be honest, it is only useful to consider RST if the author also use additional unlabeled data.
> > > >
> > > > > RST does not generalize to multiple perturbations and thus is not a competitor
> > > >
> > > > I agree, although it does get very impressive l-inf and l-2 results.
> > > >
> > > > > combined with our method
> > > >
> > > > Great new results.

---

### Author Response · Authors · 2020-11-17
**Summary of updates in the initial revision**

We thank all reviewers for their time and efforts in reviewing our paper. We also thank the reviewers for their insightful comments and constructive suggestions, which helped us to further strengthen our paper. We appreciate the positive comments from the reviewers. R1 and R3 find the work novel, R1 mentions that our work is convincing and important, and R3, R4 highlight that the paper is well-written.  Moreover, all reviewers appreciate that we provide a comprehensive set of experiments. Here we briefly mention what has been updated in the revision. **We have highlighted the updates in blue (Please see the revised version of the paper).** For more detailed explanations, please refer to the response to each reviewer.

1. **Explicit definition of AC loss:** In Section 4, based on R1's comment, we have explicitly defined the Adversarial Consistency (AC) loss in Eq. (7) of the revision.

2. **Clarification on the set of attacks and sampling distribution:** We incorporated the suggestions by R2, and explicitly defined our sampling procedure in Eq. (5) and fixed the notations for the set of attacks. Further, we elaborated Algorithm 1 and clarified the notation for the norm-ball of the attack procedure (in Algorithm 1 and SAT).

3. **Related work on generative models for adversarial robustness:** In Section 2, based on R4's comment, we included discussions on various works utilizing generative models for adversarial robustness.

4. **Intuition for the work:** We have elaborated the intuition of our learning scheme in a separate paragraph in Section 4 in the revision.

5. **Minor fixes:** We have updated the caption of Table 1, and the typo pointed out by R4.

---

### Author Response · Authors · 2020-11-18
**Author Response**

Dear Reviewers,

We sincerely appreciate your time and effort to review our paper. Since the first phase of the response period has ended, if you have time could you please indicate if there are any other concerns of yours which we have not addressed, we would be pleased to clarify those points and further strengthen our paper.

Thank you very much

---

### Decision · Program_Chairs · 2021-01-07
**Final Decision**

**Decision:**

Reject

**Comment:**

This is a borderline case. The paper seems solid although some of the numbers are likely incorrect because in some results tables in the appendix the error taken over all attacks is higher than for the best individual attack (which should never happen).

The main contribution of this paper is to augment a standard adversarial loss (against attacks from different norms) with a “consistency” term (consistency between clean, adversarial and noise augmented samples). The relatively large jump in robustness compared to existing schemes that do adversarial training against multiple norms is a bit surprising. A possible explanation could be that the additional consistency term smoothes the landscape around the clean samples a little bit, which could help to find better adversarial examples. The latter would be very similar to a paper by Pushmeet and colleagues (https://arxiv.org/pdf/1907.02610.pdf) which is not cited, but definitely should. It might also be worthwhile to compare to this paper.

Taken together, this work is interesting but not sufficiently convincing yet to belong to the top papers to be selected for publication at ICLR.